# Network of hotspot interactions cluster tau amyloid folds

Vishruth Mullapudi [1,4], Jaime Vaquer-Alicea[1,4], Vaibhav Bommareddy [1], Anthony R. Vega[1], Bryan D. Ryder [1,2], Charles L. White III [1], Marc. I. Diamond [1] & Lukasz A. Joachimiak [1,3] ✉

Cryogenic electron microscopy has revealed unprecedented molecular insight into the conformations of β-sheet-rich protein amyloids linked to neurodegenerative diseases. It remains unknown how a protein can adopt a diversity of folds and form multiple distinct fibrillar structures. Here we develop an in silico alanine scan method to estimate the relative energetic contribution of each amino acid in an amyloid assembly. We apply our method to twenty-seven ex vivo and in vitro fibril structural polymorphs of the microtubule-associated protein tau. We uncover networks of energetically important interactions involving amyloid-forming motifs that stabilize the different fibril folds. We evaluate our predictions in cellular and in vitro aggregation assays. Using a machine learning approach, we classify the structures based on residue energetics to identify distinguishing and unifying features. Our energetic profiling suggests that minimal sequence elements control the stability of tau fibrils, allowing future design of protein sequences that fold into unique structures.

Deposition of β-sheet-rich amyloids is associated with both a diverse group of neurodegenerative and systemic diseases as well as a set of functional amyloid proteins. Amyloidogenic proteins are highly variable in sequence, structure, and cellular function. These proteins assemble into highly stable fibrillar aggregates composed of layers of β-strands oriented orthogonally to the fibril axis[1,2]. Despite this common "cross-β" architecture and fibrillar morphology, amyloid fibrils adopt a diverse range of structures. In these fibrils, amyloid proteins are folded into monomer layers which are themselves stacked to form an amyloid protofilament. These protofilaments can then associate to form either single or multi-protofilament amyloid fibrils[2].

Aggregation of amyloidogenic proteins is implicated in a variety of diseases, including transmissible spongiform encephalopathies (prion protein/PrP), light chain amyloidosis (immunoglobulin light chain), Parkinson's Disease (α-synuclein/α-syn), type two diabetes (islet amyloid polypeptide/IAPP), transthyretin amyloidosis

(transthyretin/TTR), and Alzheimer's disease (amyloid-β/Aβ and microtubule-associated protein tau/tau)[3,4]. Other amyloidoses are similarly linked to various forms of neurological or systemic dysfunction. The formation and extension of these fibrils have been theorized to follow prion-like mechanisms, in which a misfolded 'seed' protein provides a template that catalyzes the misfolding and fibrillar aggregation of native-state proteins[5]. This prion-like misfolding mechanism is implicated in the transmission and propagation of amyloid pathology in the distribution of amyloid fibrils and disease progression within an organism, and (in the case of PrP) between organisms[6,7]. Prions produce stable, unique conformational 'strains', seeds of which reliably propagate both structure and patterns of disease progression. It has been demonstrated that PrP[8], Aβ[9], tau[10], and α-synuclein[11] all adopt conformations capable of propagating aggregate structure and disease pathology, However, until recently, the relation of these strains to amyloid fibril structure has been unclear.

[1]Center for Alzheimer's and Neurodegenerative Diseases, Peter O'Donnell Jr. Brain Institute, University of Texas Southwestern Medical Center, Dallas, TX 75390, USA. [2]Molecular Biophysics Graduate Program, University of Texas Southwestern Medical Center, Dallas, TX 75390, USA. [3]Department of Biochemistry, University of Texas Southwestern Medical Center, Dallas, TX 75390, USA. [4]These authors contributed equally: Vishruth Mullapudi, Jaime Vaquer-Alicea. ✉e-mail: Lukasz.Joachimiak@UTSouthwestern.edu

Tau is a key amyloid-forming protein implicated in tauopathies, a group of neurodegenerative disorders that includes Alzheimer's disease (AD), corticobasal degeneration (CBD), progressive supranuclear palsy (PSP), and others[12]. These tauopathies are characterized by the pathologic aggregation of tau into insoluble amyloid fibrils, and the presence and abundance of these fibrillar deposits are associated with dementia and reduced cognitive function[13]. Under normal conditions, tau is intrinsically disordered and soluble. Tau functions to stabilize microtubules in an extended conformation recently demonstrated in a cryo-EM structure[14]. Over the last 5 years, advances in cryo-electron microscopy (cryo-EM) and the development of helical reconstruction methods have enabled the structure determination of a variety of fibrillar conformers of tau, each associated with a particular tauopathy[15–22]. Beyond tau, several amyloid proteins including PrP[23], Aβ[24–26], IAPP[27,28], and α-syn[29–33] have also been shown to adopt multiple structurally distinct conformations, both with and without mutations. These structures have revealed unprecedented molecular details of fibrillar assemblies derived from patient and recombinant sources of tau, α-syn, TTR, IAPP, Aβ, and other proteins, but the molecular rules for how these structures are formed remain poorly understood, particularly in vivo[1,19,34–36]. At the time of publication, there have been nine distinct conformations of tau fibrils extracted from human brain tissue (16, if different protofilament arrangements are considered) thus far determined using cryo-EM, highlighting the capacity of a single protein sequence to adopt a diversity of structures associated with distinct disease phenotypes[15–22].

Both the physiological and biophysical determinants of structural polymorphism, as well as the possible range of polymorphic aggregate conformations that a single protein can adopt, are uncertain. The mechanisms by which tau adopts aggregation-prone conformations remain unclear, but tau has recently been predicted to adopt monomeric aggregation-prone 'seed' conformations via termini-stabilized rearrangements of local conformations within the microtubule-binding region[37,38]. Furthermore, pathologic monomers derived from different tauopathies have been suggested to encode the information needed to replicate the disease conformation[39] with the appearance of these species preceding the appearance of soluble oligomers and fibrils[40]. The local motifs driving tau aggregation have been proposed to be the two core amyloid-forming elements $^{275}$VQIINK$^{280}$ and $^{306}$VQIVYK$^{311}$ located at the beginning of repeat domains two and three, respectively[37]. These amyloid motifs are predicted to be normally engaged in local interactions within tau to limit their aggregation propensity[41]. Recent analysis of cryo-EM fibril structures—including tau fibrils—highlights the possibility that amyloid motifs may play important roles in stabilizing the distinct folds, suggesting that inert forms of tau must rearrange structurally to adopt pathogenic conformations[42,43]. Derived from this work, local structural rearrangements surrounding the amyloid motifs encoded in a tau monomer have been proposed to drive differentiation into distinct structural polymorphs[39]. Thus, it is possible that specific misfolding events in a monomer are sufficient to initiate the assembly of tau into conformationally distinct aggregates.

To explore the determinants of amyloid polymorphism, we use tau as a model protein to understand how different tau fibril folds may form and what interactions may mediate their stability. We note that amyloidogenic motifs in tau play important roles in stabilizing heterotypic nonpolar contacts within tau fibrils. To further understand the interactions responsible for stabilizing amyloid fibrils, we deploy an in silico method using Rosetta to probe residue energetics in across different fibrillar structures. We first develop a minimization protocol for fibrils yielding minimized structures that retain near native backbone conformations and recapitulate side chain rotamers and interactions. Using this platform, we implement an alanine mutagenesis scan for 27 ex vivo and recombinant tau fibril structures and estimate the relative energetic contribution of specific residues to the stability of tau fibrils. We use these estimated contributions to identify the thermodynamic hotspots that contribute to and differentiate the multiplicity of fibril polymorphs found in tauopathies. We uncover key hotspot residues involving amyloidogenic motifs and identify a modular network of interactions of which subsets are preserved across structurally diverse folds. We proceed to test the role of these heterotypic interactions in peptide co-aggregation and in cell experiments and show that amyloid motif-dependent aggregation can be regulated by heterotypic contacts. Finally, we leverage a machine learning approach to uncover key energetic features that will help classify the different structures.

## Results

### Tauopathy fibril cores involve modular interactions with amyloid motifs

To understand how primary sequence properties may underpin tau's amyloid polymorphism, we compared tau's polar and nonpolar amino acid distribution with its predicted aggregation propensity. Aggregation-promoting regions (APRs) of tau have previously been identified, and prior work suggests sequences surrounding these APRs may regulate their contribution to tau self-assembly[41,44]. Structure-based computational methods (e.g., Pasta, ZipperDB, Waltz) can predict aggregation-promoting elements and have uncovered key motifs including $^{275}$VQIINK$^{280}$ and $^{306}$VQIVYK$^{311}$ as APRs central to tau aggregation (Supplementary Fig. 1a). Other sequences that are predicted to be only weakly amyloidogenic retain increased hydropathy (Fig. 1a) and may be involved in heterotypic interactions with core amyloid motifs[45–50]. Indeed, $^{275}$VQIINK$^{280}$ and $^{306}$VQIVYK$^{311}$ aggregate into ordered fibrils, but other sequences predicted by ZipperDB to be close to the energetic threshold (Fig. 1a, dashed red line) form a distribution of ordered and disordered (i.e., amorphous) aggregates (Fig. 1b). Sequences close to or below this threshold remain soluble (Supplementary Fig. 1b) and do not yield Thioflavin T (ThT) fluorescence signal, a reporter of ordered β-sheet structure formation (Supplementary Fig. 1c). Focusing on the well-characterized $^{306}$VQIVYK$^{311}$ peptide, we wanted to understand which residues in this motif are important for fibril formation. To test this directly, we measured the aggregation capacity of $^{306}$VQIVYK$^{311}$ peptide mutants substituted with alanine at each position in a ThT fluorescence aggregation assay. We find that alanine mutations at positions one, three, and four (i.e., **A**QIVYK, VQ**A**VYK, and VQI**A**YK) completely abolish ThT fluorescent signal while substitutions at positions two and six (i.e., V**A**IVYK and VQIVY**A**) retain near WT aggregation properties (Supplementary Fig. 1d). Position five (i.e., VQIV**A**K) gives intermediate fluorescence signal (Supplementary Fig. 1d), but this may be due either to the loss of the aromatic residue which is important for ThT binding or to a decrease in fibrilization[51]. We confirmed our ThT results by imaging each sample using negative stain transmission electron microscopy (TEM) and find that the peptides that yield positive ThT signal have fibrils. One exception is the **A**QIVYK peptide, which yielded thin fibrils that were negative in the ThT assay, suggesting that this mutation alters the morphology of the aggregates compared to the other ThT positive structures (Supplementary Fig. 1e). To interpret these data in the context of existing structures of $^{306}$VQIVYK$^{311}$ and related sequences determined by X-ray crystallography, we find that valine and isoleucine at positions 306 (i.e., **V**QIVYK) and 308 (i.e., VQ**I**VYK) of VQIVYK appear to be central in stabilizing intermolecular contacts in either a head-on or offset register (Supplementary Fig. 1f). Additionally, valine 309 (i.e., VQI**V**YK) is important despite not forming the central hydrophobic core indicating that additional secondary interactions on both hydrophobic sides of the β-sheet are important for fibril formation. These data suggest that $^{306}$VQIVYK$^{311}$ has the capacity to stabilize self-interactions in different arrangements in simple peptides using both sides of the β-sheet. Unsurprisingly, if we compare the placement of $^{306}$VQIVYK$^{311}$ in the nine unique ex vivo tau protofilament

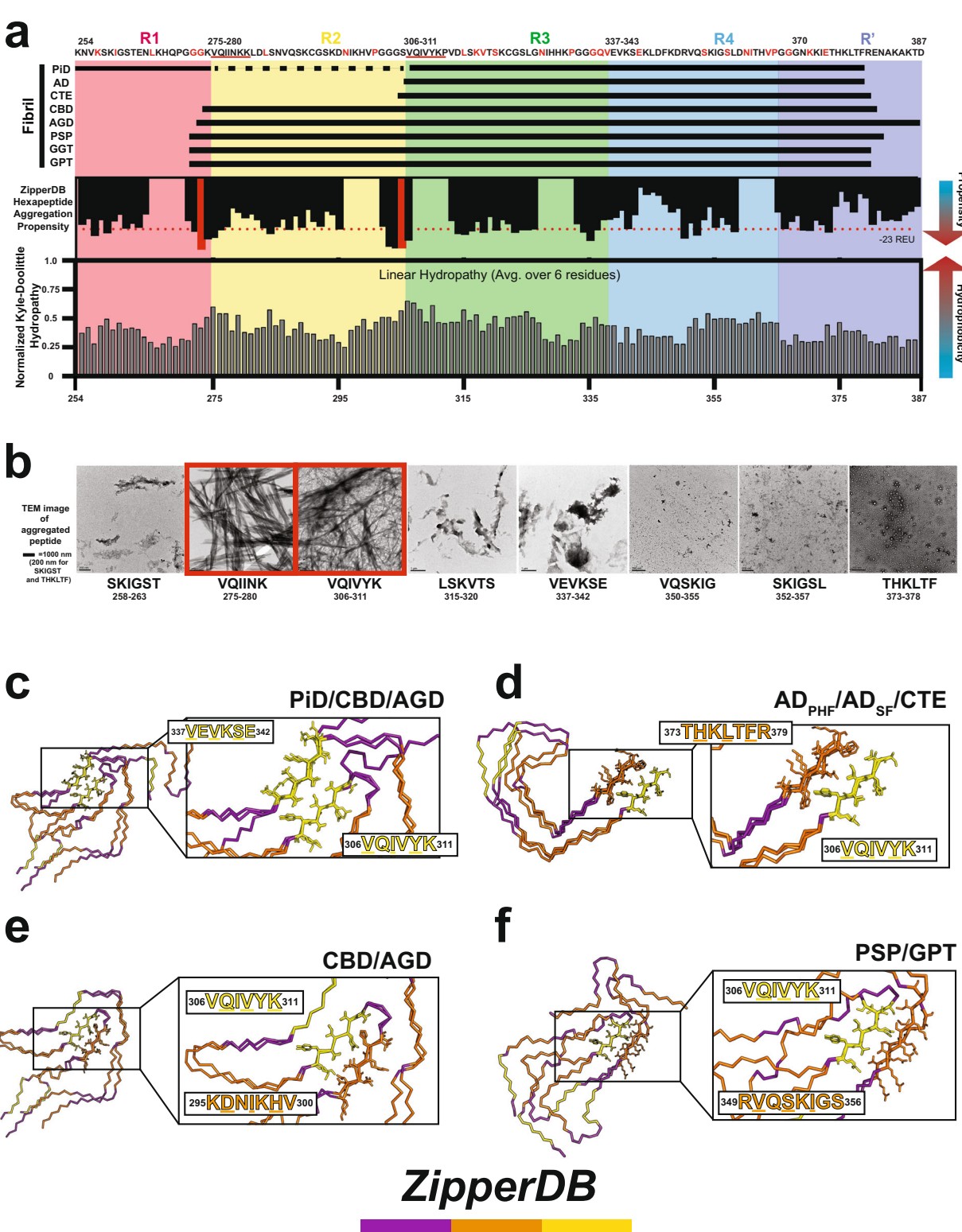

ZipperDB

non agg.        agg.

polymorphs determined to date (interpreting only a single layer), we observe that [306]VQIVYK[311] is engaged in interactions with different sequences from repeat domains two and four (Supplementary Fig. 2a) that are often aggregation-prone and match the nonpolar composition of this amyloid motif (Supplementary Fig. 2b, c)[15,17,18,20,21].

To look at the specific conservation of the interactions with [306]VQIVYK[311] we aligned a layer from each of the nine prototypical ex

vivo-derived structural polymorphs on the [306]VQIVYK[311] amyloid motif. We find that [306]VQIVYK[311] makes conserved interactions with other predicted amyloid motifs in subsets of the fibrillar structures (Fig. 1d–f). In the PiD, CBD, and AGD structures, V306, I308, and Y310 from VQIVYK make identical interactions to V337 and V339 from [337]VEVKSE[342] (Fig. 1d). In the simpler AD-PHF, AD-SF, and CTE folds, V306, I308, and Y310 from [306]VQIVYK[311] instead interact with L376 and

**Fig. 1 | Nonpolar and amyloidogenic core fragments are modular in tau fibrils.**
**a** Diagram of the tau fragments utilized in the fibrillar assemblies highlighting the aggregation propensity of the tau sequence (as estimated by ZipperDB) and its hydrophobicity (shown as Kyte−Doolittle hydropathy, calculated by localCider)[48,49,88]. The fragments for each structural polymorph are shown as horizontal bars. The tau repeat domains are colored red, yellow, green, blue, and navy blue for R1, R2, R3, R4, and R'. The corresponding tau sequence is shown on the top. Known mutations are colored in red and derived from Alzforum[76]. Key amyloid motifs are underlined in red. Aggregation propensity is shown as black bars. Dotted red line denotes the threshold for high aggregation propensity. The two most highly aggregation-prone sequences, $^{275}$VQIINK$^{280}$ and $^{306}$VQIVYK$^{311}$ are highlighted in red. Hydropathy plot is shown as gray bars. **b** TEM images of predicted amyloid motifs. $^{306}$VQIVYK$^{311}$ and $^{275}$VQIINK$^{280}$ are highly aggregation-prone (boxed in red), and form ordered assemblies while the other sequences are predicted to have lower aggregation propensity form disordered or amorphous aggregates.

Imaging of the peptide aggregates using TEM was performed two independent times. **c**–**f** Comparison of $^{306}$VQIVYK$^{311}$ amyloid motif interactions across structural polymorphs. **c** $^{306}$VQIVYK$^{311}$ forms conserved nonpolar interactions with $^{337}$VEVKSE$^{342}$ in Picks disease (PiD), Corticobasal degeneration (CBD), and Argyrophilic grain disease (AGD) structures. **d** $^{306}$VQIVYK$^{311}$ forms conserved nonpolar interactions with $^{373}$THKLTFR$^{379}$ in Alzheimer's disease-Paired Helical Filaments (AD-PHF), Alzheimer's disease-straight filaments (AD-SF) and chronic traumatic encephalopathy (CTE) structures. **e** $^{306}$VQIVYK$^{311}$ forms nonpolar interactions with $^{294}$KDNIKHV$^{300}$ in AGD and CBD structures. **f** $^{306}$VQIVYK$^{311}$ forms nonpolar interactions with $^{349}$RVQSKIGS$^{356}$ in Progressive supranuclear palsy (PSP) and globular glial tauopathy (GPT) structures. The structures are shown as a single layer in a ribbon representation and are colored by amyloidogenic propensity: yellow (high), orange (medium), and purple (low). Amino acid sequences of interacting motifs are colored by aggregation propensity from ZipperDB and amino acids important for interaction are underlined.

F378 from $^{373}$THKLTFR$^{379}$ (Fig. 1e). In the CBD and AGD folds, V306, I308, Y310, and K311 from $^{306}$VQIVYK$^{311}$ interact with D295, I297, and H299 of $^{295}$DNIKHV$^{300}$ highlighted by nonpolar contacts between V306 and I297 and a buried salt bridge between K311 and D295 (Fig. 1f). Interestingly, in GGT, I297 from $^{295}$DNIKHV$^{300}$ interacts with $^{306}$VQIVYK$^{311}$ but the other residues are out of register. Finally, in the PSP and GPT structures, Q307, V309, and K311 of $^{306}$VQIVYK$^{311}$ interact with V350, S352, and I354 of $^{349}$RVQSKIGS$^{355}$ (Fig. 1g). These analyses suggest that aggregation-prone elements including $^{306}$VQIVYK$^{311}$ are used modularly to bury nonpolar contacts in the cores of ex vivo-derived tau fibril structures. Our data indicate that the interactions of key amyloid sequences with other hydrophobic sequence elements play important roles in tau amyloid assembly and the heterogeneity of these possible stabilizing interactions may be central to the formation of a diversity of structural polymorphs. Importantly, we predict that $^{306}$VQIVYK$^{311}$ is a key regulator of tau assembly which uses one or two surfaces to stabilize hydrophobic interactions in simple and complex fibril cores. The fibrils can be classified into two general categories: one where the $^{306}$VQIVYK$^{311}$ peptide strand interacts with a second β-strand in a one-sided β-sheet interaction, and another in which $^{306}$VQIVYK$^{311}$ engages two other β-strands in a two-sided β-sheet interaction. Under this schema, we classify CBD, AGD, PSP, GGT, and GPT as fibrils with two-sided $^{306}$VQIVYK$^{311}$ interactions, while AD, CTE and PiD (and the heparin-derived fibril structures) are comprised of monomer layers with one-sided $^{306}$VQIVYK$^{311}$ interactions.

To gain a more coarse-grained view of the types of residues that are buried in the nine prototypical ex vivo structures, we colored each structure by amino acid polarity. We find that nonpolar amino acids are often buried in the fibril cores (Supplementary Fig. 2c, yellow spheres) while basic residues are presented on the outside of the fibrils (Supplementary Fig. 2c, blue spheres). We also quantified the degree of burial of different amino acid types by calculating the change in solvent-accessible surface area between a fully extended monomer to its folded monomer conformation alone ($\Delta SASA_{single\ layer}^{folding}$), or to its folded monomer in the context of a fibril ($\Delta SASA_{fibril}^{folding}$). Aggregating this data over the nine ex vivo prototypical fibril polymorphs, we find that nonpolar amino acids have both a large $\Delta SASA_{single\ layer}^{folding}$ and $\Delta SASA_{fibril}^{folding}$ (Supplementary Fig. 3a, orange), consistent with nonpolar residue burial upon tau monomer folding into a fibril conformation, and further burial as the additional layers of the fibril sandwich the initial layer. Polar and acidic residues appear to be distributed more evenly between the core and the surface (Supplementary Fig. 2c) and show similar burial patterns by $\Delta SASA_{single\ layer}^{folding}$ and $\Delta SASA_{fibril}^{folding}$ (Supplementary Fig. 3a, green and red). Interestingly, when we quantify the change in solvent accessibility for basic residues (mostly lysines) we find that $\Delta SASA_{single\ layer}^{folding}$ is small but $\Delta SASA_{fibril}^{folding}$ is large, suggesting

that basic residues are solvent exposed but can bury their side chains in the fibril by stacking their aliphatic side chains (Supplementary Figs. 2c, 3a, blue). Consistent with this observation a recent study showed that a tau fragment can assemble into AD-PHF and CTE conformations using specific buffer and salt combinations indicating that cation and anion interactions with tau may promote tau folding by stabilizing nonpolar burial and screening electrostatic interactions[52]. This analysis indicates that nonpolar residues are generally buried in tau fibril structures−often through interactions with amyloid motifs−likely contributing to the stability of the different protofilament folds.

## Nine-layer fibril stack retains native-like properties through minimization and mutagenesis

We developed a framework to interpret the interaction energies across a panel of fibrillar structures using the Rosetta software package[53]. Rosetta was originally developed to predict structures of proteins from the sequence with applications expanded to protein−protein and protein−ligand docking, protein design, and RNA folding[54–57]. At its core, Rosetta employs a parametrized energy function that combines the energetics of different terms to capture physics-based interactions[58]. This energy function is combined with structure-derived fragments and a Metropolis Monte Carlo sampling approach to build structural models that resemble experimentally determined protein structures. We first used Rosetta's parametrized energy function to minimize each fibril structure and then applied a Flex-ddG-based protocol to perform alanine-scanning mutagenesis which allows us to infer the energetic contribution of amino acids in each structure (Fig. 2)[59]. Additionally, implementation of the backrub protocol is used to improve sampling of the backbone and side chain conformational space and improve energy minimization (Fig. 2)[60]. Alanine scanning mutagenesis has a long history and has been a key experimental and computational tool to infer residue contributions to protein folding and protein interactions[61–66]. While mutations to alanine can alter the backbone torsional angles, they effectively approximate the contribution of a side chain to the stability of intramolecular and intermolecular interactions.

Unlike globular proteins, which can exist stably in monomeric or a set of defined oligomeric states, fibrils exist as a set of layered monomers without a specific required length/number of layers. To determine a sufficient number of monomer layers for use in the fibril simulations, we minimized the AD-PHF and CBD structures using different numbers of monomer layers from a trimer to a nine-mer representing the fibril (Fig. 3a). Minimization yielded energetically favorable assemblies compared to the starting fibril structures with the total energy of the assembly scaling with the number of layers for both wild-type and alanine mutants (Supplementary Fig. 3b, c). Additionally, we computed RMSD distributions for AD-PHF and observe that the RMSD decreases with increasing numbers of layers (Fig. 3b, blue) while for CBD the RMSD remains flat across the range of layers (Fig. 3c, blue)

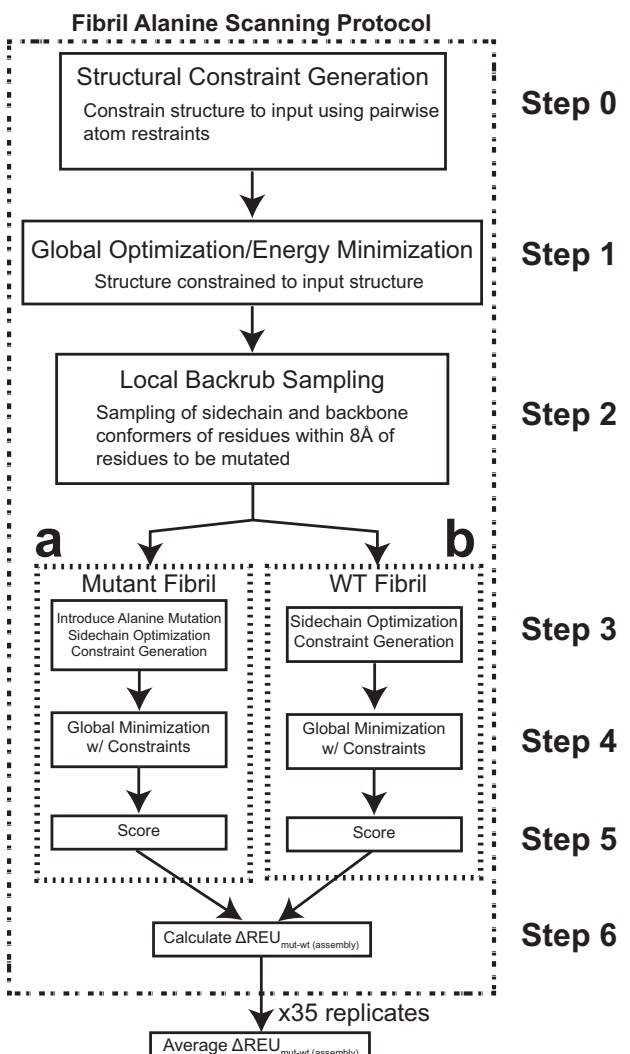

**Fibril Alanine Scanning Protocol**

**Fig. 2 | Diagram for the fibril Flex ddG energy estimation method.** Outline of fibril flex $\Delta REU^{assembly}_{mut-wt}$ protocol. (0) The fibril is constrained to the input fibril structure with atom pair constraints. (1) The input fibril structure is minimized using harmonic restraints until convergence (absolute score change upon minimization of less than one Rosetta Energy Unit (REU)). (2) The backrub method[59] is used to sample additional backbone and side chain conformers proximal to the mutation site. Each backrub move is undertaken on a randomly chosen protein segment consisting of three to 12 adjacent residues with a C-β atom (C-α for glycines) within 8 Å of the mutant positions, or adjacent residues. All atoms in the backrub segment are rotated locally about an axis defined as the vector between the endpoint C-α atoms. Backrub is run at 1.2 kT, for 35,000 backrub Monte Carlo steps. 35 ensemble models are generated. (3A) Alanine mutants are introduced to the backrub-sampled model, and side chain conformations for the mutant structure are optimized using the Rosetta packer. (3B) alongside the alanine mutant model, a wild-type model is also optimized with the packer. (4A) The mutant model is minimized using pairwise atom-atom constraints to the input structure. Minimization is run with the same parameters as in step 1; the coordinate constraints used in this step are taken from the coordinates of the step 3A model. (4B) Same as step 4A, but for the wild-type model. (5A) The model is scored both as a 'bound' protein complex and as a split, 'unbound' complex. The scores of the split, unbound complex partners are obtained simply by moving the complex halves away from each other. No further minimization or side-chain optimization is performed on the unbound partners before scoring. (5B) similarly to step 5A, the minimized wild-type model is scored as both in a 'bound' and 'unbound' state (6). The interface $\Delta\Delta G$ and $\Delta REU^{assembly}_{mut-wt}$ score using the following equations (1):

$$\Delta\Delta G^{interface} = [\text{bound} - \text{unbound}]_{mut} - [\text{bound} - \text{unbound}]_{wt}$$
$$\Delta REU^{assembly}_{mut-wt} = \text{bound}_{mut} - \text{bound}_{wt}$$

These $\Delta\Delta G^{interface}$ and $\Delta REU^{assembly}_{mut-wt}$ values are then averaged over the 35 replicates in the ensemble to generate the final $\Delta\Delta G$ and $\Delta REU^{assembly}_{mut-wt}$ for the residue.

for both wild-type and alanine-substituted structures. We suspect the number of layers may help retain the assembly in a native conformation and may help with fibrils with simpler topologies (i.e., improved AD more than CBD). Comparing the minimized nine-mer AD-PHF and CBD structures to the native conformation, we find that our minimization protocol maintains near native backbone and side chain rotamers in the AD and CBD structures with all-atom RMSDs of 0.486 and 0.539 Å, respectively (Fig. 3d, e, blue). Importantly, we recover the native rotamers of side chains in the core of the fibril, though residues with side chains on the outside surface that mainly interact with the solvent are more variable (Fig. 3d, e, blue). For more in-depth analysis across tau fibrils, we decided to move forward with nine-mer assemblies which balanced low RMSD with the cost of computation time.

Using the nine-layer fibril framework, we minimized 27 structures covering all known tau fibril structural polymorphs, including both patient-derived and recombinant fibrils, yielding structures that have near-native RMSDs (Fig. 3f). The starting fibril structures overall are not energetically favorable when scored with Rosetta but following iterations of minimization, we produce low energy conformations for wild-type and all alanine mutant structures (Supplementary Fig. 3d). When comparing the wild-type and alanine mutant structures generated, we find that overall, the alanine mutants are only slightly destabilizing, suggesting that alanine substitutions do not substantially alter the overall energy distributions when considering them cumulatively across all positions (Fig. 3f) or induce major deviation from the native conformation (Supplementary Fig. 3d). We find that the fibril

structures with simpler topologies, namely the AD-PHF, AD-SF, and heparin-derived fibrils with only one-sided [306]VQIVYK[311] interactions, yield energies between −700 REU and −1200 REU while the more complex topologies (CBD, AGD, PSP, GPT and GGT) tend to yield lower energies around −1300 to −1600 REU (Supplementary Fig. 3d, bottom panel). Interestingly, the structures of recombinant fibrils induced with heparin yielded the highest energies (i.e., least stable structures (Supplementary Fig. 3d, bottom panel). To explain this in more detail, we normalized the energies to the number residues in the assembly (Supplementary Fig. 3d, top panel) and find that indeed the recombinant heparin-derived structures have some of the highest per residue energies. This may be explained by either overall poorer packing of the residues in the cores of the heparin-derived tau fibrils or the possibility that these structures are stabilized by heparin binding on the surface. Interestingly, while the AD-PHF/AD-SF structures are similar to the CTE structures, we find that the CTE structures have equal or lower per residue contribution suggesting that the more open "C" conformation may be better packed than the more closed "C" from AD. Similarly, the other two-layer PiD structure has very low per-residue energetics. Of the complex three-layer fibril topologies, PSP, GGT and GPT have the lowest per residue energies compared to AGD and CBD suggesting there is large variation in the side chain energetics of the more complex folds. Two of the CBD structures (PDB IDs 6vh7 and 6vha) did not minimize as well, leading to higher per-residue energetics. This may be due to lower structure resolution causing slight conformational differences from the higher resolution, better minimizing CBD structures (PDB IDs 6tjo and 6tjx). Finally, related to the stabilization of the heparin fibrils by ligands, the cryo-EM maps of CTE and CBD fibrils showed unexplained densities whose interactions with the protein sequence likely yield additional stabilization[17,18]. We next evaluated in more detail whether alanine substitutions alter the fibrillar conformation and influence the energy of the structures. We find that in our backrub sampling protocol followed by alanine substitutions and minimization does not significantly alter the overall RMSD distributions in the structures as compared to the native conformation (Fig. 3b, c). For example, the conformations of the minimized AD PHF

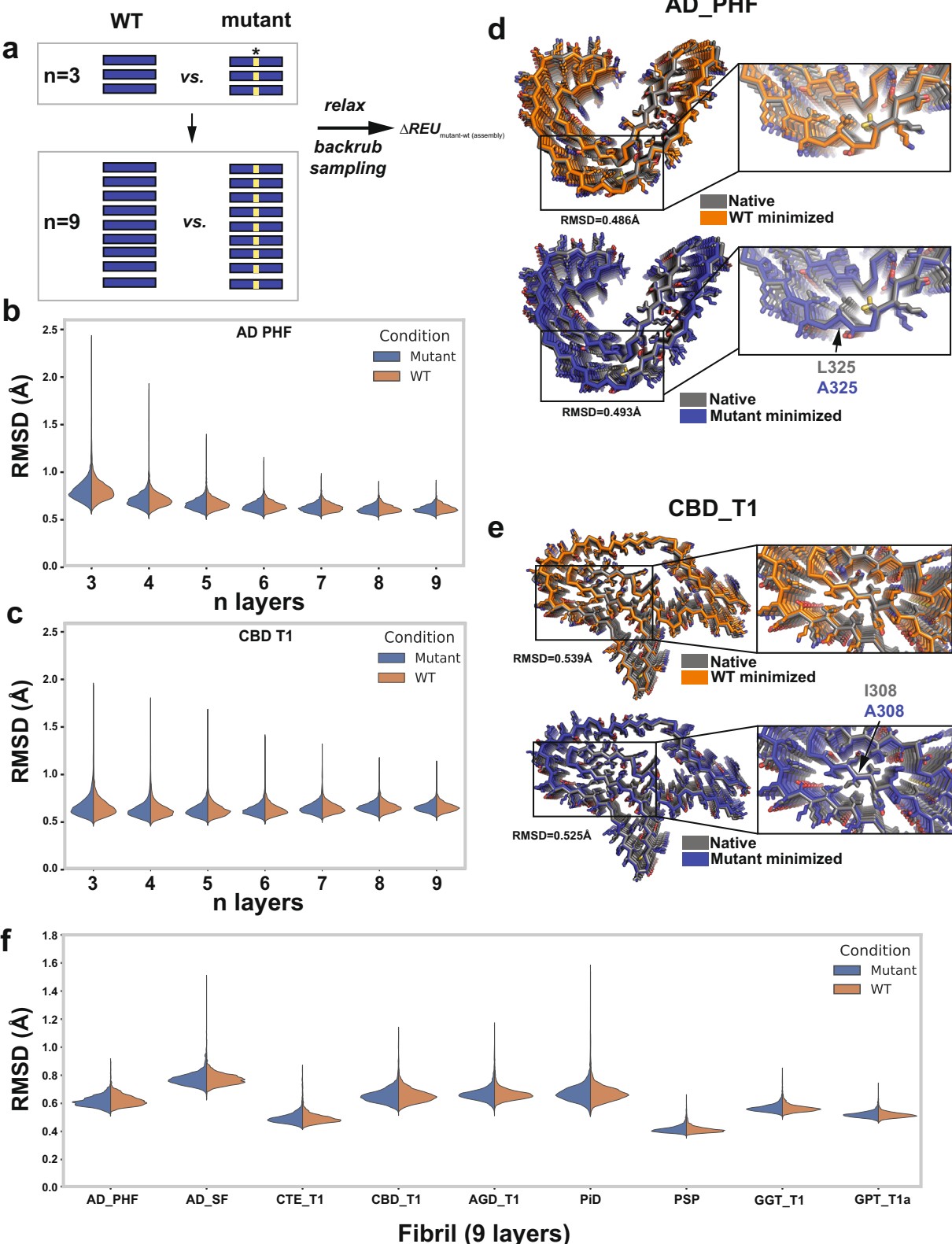

L325A and the CBD I308A remain similar to the native structure with RMSDs of 0.493 and 0.525 Å, respectively (Fig. 3d, e).

We found, however, that glycine to alanine substitutions often resulted in anomalously high structural energies in the AD-PHF and CBD structures, with the perturbations scaling with the number of layers (Supplementary Fig. 3e, f). We note that glycine residues in all deposited (35 total) tau fibril structures (red points, Supplementary Fig. 3g) frequently adopt alternate torsional angles that are not often sampled in globular proteins (X-ray structures between 1.5 and 2.5 Å resolution). This observation is further extended to all deposited fibril structures (96) determined by cryo-EM using helical reconstruction in Relion (Supplementary Fig. 3h). We are unsure whether this is because the cryo-EM density maps are underdetermined and the energy of the model fitting is driven by empirical terms from force fields or whether

**Fig. 3 | Fibril alanine scanning method produces minimized fibril assemblies that retain native contacts. a** Schematic illustrating Flex ddG-based protocol to minimize and substitute alanine at individual positions in amyloid fibrils using different numbers of monomer layers. Blue box highlights a single layer. Yellow box highlighted by star denotes location of mutation. **b** and **c** Root mean square deviation (RMSD) distributions for WT (orange) and alanine mutants (blue) across different numbers of layers ($n = 3$ to $n = 9$) for AD-PHF (PDB id: 5o3l) (**b**) and CBD (PDB id: 6tjo) (**c**). **d** Structural overlay of AD-PHF nine-layer native WT (gray) and minimized WT (orange) reveals a RMSD of 0.486 Å (top). Structural overlay of AD-PHF nine-layer native WT (gray) and minimized L325A mutant (orange) reveals a RMSD of 0.493 Å (bottom). **e** Structural overlay of CBD nine-layer native WT (gray) and minimized WT (orange) reveals a RMSD of 0.539 Å (top). Structural overlay of CBD nine-layer native WT (gray) and minimized I308A mutant (orange) reveals a RMSD of 0.525 Å (bottom). **f** RMSD distributions for minimized WT (orange) and alanine mutant (blue) nine-layer structures reveal low RMSDs that range from 0.3 to 0.6 Å. The RMSD distributions are shown across 35 replicates at each position in each fibril and plotted as violin plots. AD-PHF, CBD_T1, CTE_T1, PiD, AGD_T1, PSP, GGT_T1a, and GPT_T1 for PDB ids: 5o3l, 5o3t, 6gx5, 6nwp, 6tjo, 7p6d, 7p65, 7p66, and 7p6a.

these are bona fide features of the fibril structures. To see whether the backbone torsional angles for glycine residues are compatible with alanine substitutions we compared the glycine residue distributions in the tau fibril structures to observed alanine torsional angles in our globular protein dataset and find that they often are not permissible (Supplementary Fig. 3i).

As such, the substitution of glycine residues in the fibril structures with alanine often yields large changes in energy dominated by the full atom Lennard–Jones repulsive potential energy component suggesting the formation of atom clashes with a minor effect from the rotamer energy term (Supplementary Figs. 3e, f, 4a; fa_rep, fa_dun). Consistent with our original observations glycine to alanine substitutions contribute large increases to the repulsive potential energy component likely through the formation of atom clashes but consequently also contribute favorably to the attractive potential energy term (Supplementary Fig. 4a; fa_rep, fa_atr). Between this deviation in the backbone torsional angles and the energetic consequences of introducing atom clashes, we justify the exclusion of glycine to alanine mutants from the analysis of energies. Excepting these glycines to alanine mutants, our protocol appears to yield energetically favorable structures without large conformational deviations from the native.

As the nine-layer monomer stack preserves the native conformation in our protocol, we can investigate the native state interactions found in tau fibrils. Using mutagenesis as a probe, we proceeded to in-silico energetic analysis of 27 patient-derived and recombinant structures of tau fibrils[15–18,20,21].

## In silico alanine-scanning captures the energetics of native fibril conformations

The Flex-ddG framework was originally implemented to capture the changes in binding energies for protein complexes[59]. To adapt the Flex-ddG method to our fibrillar system we attempted two approaches. First, we probed the inter-layer energetics of the fibril by separating a central trimer from the rest of the nine-mer as the "unbound state" and compared its energetics to a "bound" state, where the central trimer occupies its native position in the fibril in order to probe the interface energy between layers of the fibril (Supplementary Fig. 4b). We find that the predicted $\Delta\Delta G^{\text{interface}}$ (i.e., the $\Delta\Delta G_{\text{mut−wt}}^{\text{binding}}$ of the central trimer interacting with the remainder of the nine-layer fibril) has relatively small changes in response to alanine substitutions (Supplementary Fig. 4c, top panel). This suggests that the surfaces between the fibril monomers are relatively flat and do not have features that significantly stabilize the interactions beyond the backbone hydrogen bonding pattern which scales with the number of amino acids and that additional edge layers expose a higher percentage of nonpolar amino acids which contribute negatively to solvation energies.

As the $\Delta\Delta G^{\text{interface}}$ does not capture the energetic changes of mutation within layers of the fibril, only between layers, we next compared the difference in total energies of nine-mer assemblies between wild-type and alanine mutants (i.e., the $\Delta\text{REU}_{\text{mut−wt}}^{\text{assembly}}$), interpreting the total energy of the nine-mer assemblies and observe perturbations that are ten-fold greater than the "bound" vs. "unbound" interface energies (Supplementary Fig. 4c, bottom panel) and correlate with each other across the nine prototype structural polymorphs

(Supplementary Fig. 4d). Although our approach cannot accurately estimate the energy of the unfolded ensemble of tau to predict a true $\Delta\Delta G^{\text{mutation}}$, this approach captures changes in both inter-layer and intra-layer energetics resulting from mutation, allowing us to interpret the contribution of an individual residue to the stability of the amyloid fibril. We additionally evaluated the contribution of edge layers to the entire assembly by comparing the energetics of edge layers ($\Delta\text{REU}_{\text{mut−wt}}^{\text{edge}}$), internal layers ($\Delta\text{REU}_{\text{mut−wt}}^{\text{internal}}$) to the energetics of the entire assembly ($\Delta\text{REU}_{\text{mut−wt}}^{\text{assembly}}$) for AD-PHF and CBD across numbers of modeled layers (Supplementary Fig. 4e, $n = 3–9$). As described above, we excluded contributions of alanine substitutions at glycine positions due to backbone torsional and steric clash considerations. We find that the energetics of $\Delta\text{REU}_{\text{mut−wt}}^{\text{assembly}}$ and $\Delta\text{REU}_{\text{mut−wt}}^{\text{internal}}$ substitutions to alanine both destabilize the fibrils and that these values scale with the number of layers for AD-PHF and CBD assemblies (Supplementary Fig. 4f, g, top and middle panels). In contrast, we find the edge layers improve the energetics, albeit marginally, suggesting that mutation to alanine on the edges is stabilizing (Supplementary Fig. 4f, g, bottom panel) likely due to improved solvation energies for nonpolar residues. We also find that the $\Delta\text{REU}_{\text{mut−wt}}^{\text{internal}}$ and $\Delta\text{REU}_{\text{mut−wt}}^{\text{assembly}}$ energies correlate positively (Supplementary Fig. 4h) while $\Delta\text{REU}_{\text{mut−wt}}^{\text{internal}}$ and $\Delta\text{REU}_{\text{mut−wt}}^{\text{edge}}$ energetics correlate negatively or not at all (Supplementary Fig. 4i). Additionally, the correlation of the $\Delta\text{REU}_{\text{mut−wt}}^{\text{internal}}$ and $\Delta\text{REU}_{\text{mut−wt}}^{\text{edge}}$ energetics decreases as a function of layers, from directly correlated to anticorrelated, likely as the internal layers become more unlike the edge chains due to increased fibril length. Computing the coefficient of correlation between the $\Delta\text{REU}_{\text{mut−wt}}^{\text{assembly}}$ and $\Delta\text{REU}_{\text{mut−wt}}^{\text{internal}}$ datasets, shows correlations that range from 0.94−0.96 and 0.91−0.94 for the AD-PHF and CBD datasets, respectively, and are independent of a number of layers while $\Delta\text{REU}_{\text{mut−wt}}^{\text{edge}}$ and $\Delta\text{REU}_{\text{mut−wt}}^{\text{internal}}$ datasets correlate less well and shift to a negative correlation with a larger number of layers (Supplementary Fig. 4j, k). This emphasizes the need to minimize larger assemblies where the energetic distinction between internal and edge layers is greater. Importantly, the top destabilizing positions when mutated to alanine identified in the $\Delta\text{REU}_{\text{mut−wt}}^{\text{assembly}}$ overlap with hits detected with energetics derived from only $\Delta\text{REU}_{\text{mut−wt}}^{\text{internal}}$ (Supplementary Fig. 4h). Our analysis suggests that calculation of $\Delta\text{REU}_{\text{mut−wt}}^{\text{assembly}}$ with larger numbers of layers in the assembly yields structures with lower RMSDs and scores yielding better good signal to noise at identifying stabilizing sites.

We proceeded to use this second, $\Delta\text{REU}_{\text{mut−wt}}^{\text{assembly}}$ method for the analysis of 27 tau structures using the nine-layer assembly format, systematically substituting alanine residues at each position in the nine-layer stack and comparing the total energy to that of the wild-type assembly (Fig. 3a). From the "total energy" change between the wild-type and mutant assemblies, looking at the 9 distinct ex vivo tau protofilament structures we find that there is an alternating pattern consistent with β-sheets where residues facing inward contribute more than residues facing outward (i.e., solvent-facing residues) (Supplementary Fig. 4c, bottom panel). We parsed the different energy terms to understand the origins of the energy differences. On inspection of the energy changes for different amino acid types upon mutation to alanine, we find that non-polar amino acids (Ile, Val, Leu, Phe, Tyr, and Met) have the largest loss in the attractive potential energy term and

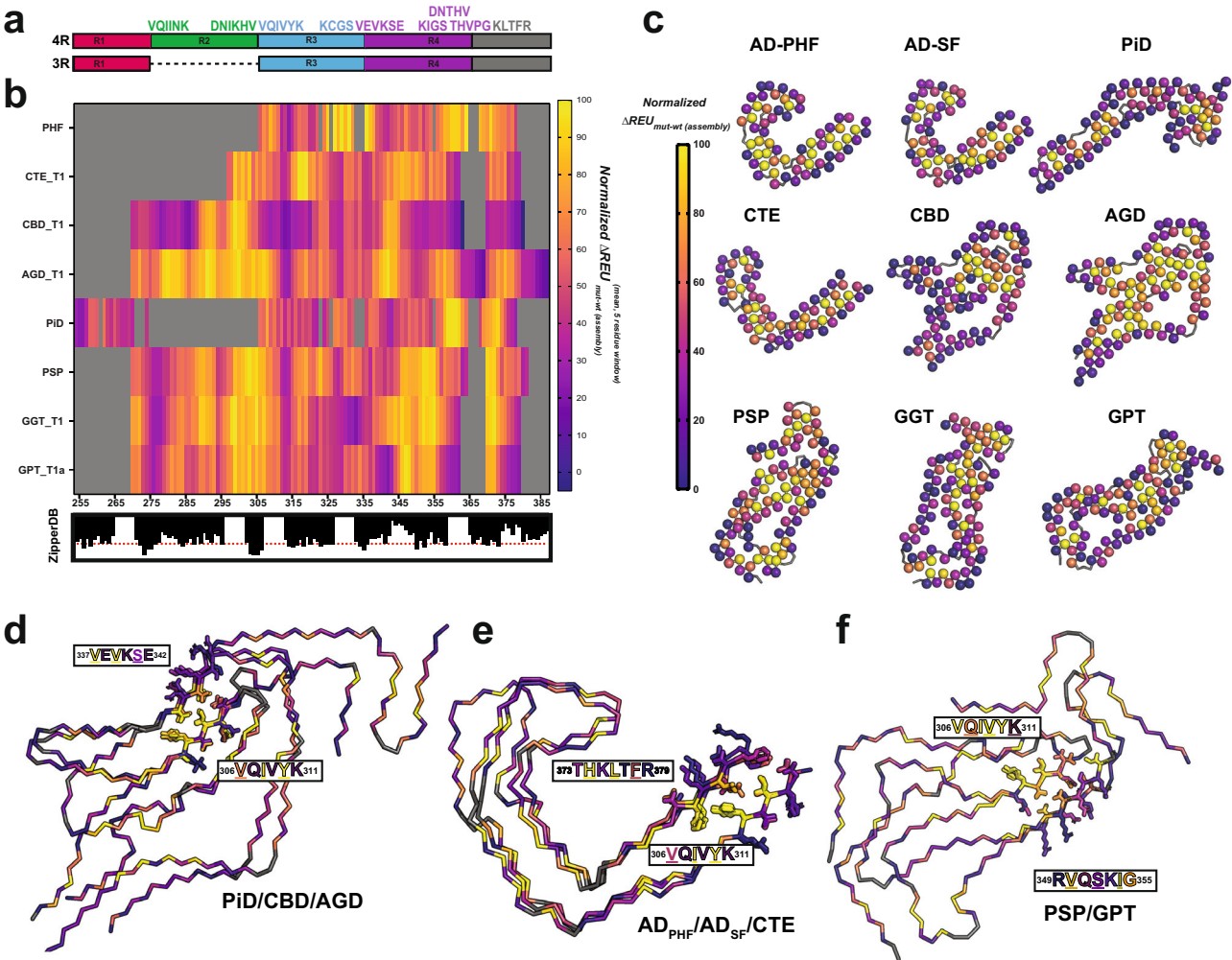

**Fig. 4 | Fibril alanine scanning uncovers modular networks of hotspot interactions in tau fibrils. a** Cartoon illustration of 4R and 3R isoforms of the tau repeat domains colored as in Fig. 1a highlighting location of key amyloidogenic motifs shown above. **b** Normalized heatmap of energetic change in response to substitution to alanine across a 5-residue window for 9 prototypical tau fibril structures: AD-PHF, CBD_T1, CTE_T1, PiD, AGD_T1, PSP, GGT_T1a, and GPT_T1. PDB ids: 5o3l, 6gx5, 6nwp, 6tjo, 7p6d, 7p65, 7p66, and 7p6a. Scale is colored in plasma from yellow (important) to purple (not important). **c** Mapping per residue $\Delta REU^{assembly}_{mut-wt}$ profiles onto the 9 tau fibril structures uncovers inward-facing amino acids contribute more to stability than surface exposed residues. Structures are shown as a single layer shown in ribbons with C-α atoms shown in spheres for each amino acid. The C-β is colored by changes in energy due to individual amino acid mutations to alanine using the plasma color scheme, yellow (100 normalized ΔREUs, important) to purple (0 normalized ΔREUs, not important). **d–f** Structural overall of similar structures highlighting energetically important interactions with VQIVYK. **d** Overlay of PiD, CBD, and AGD highlighting the energetically important and conserved interactions between [306]VQIVYK[311] and [337]VEVKSE[342]. **e** Overlay of AD-PHF, AD-SF, and CTE highlighting the energetically important and conserved interactions between [306]VQIVYK[311] and [373]THKLTFR[379]. **f** Overlay of PSP and GPT highlighting the energetically important and conserved interactions between [306]VQIVYK[311] and [337]VEVKSE[342]. The fibril structures are shown as a single layer in ribbon representation, the interacting motifs are shown as sticks and colored by their normalized per residue energy contribution using the plasma color scheme (yellow = 100 and purple = 0).

some gain in the repulsive potential energy term, consistent with creating destabilizing cavities (Supplementary Fig. 4a; fa_atr, fa_rep) in the cores of fibrils (Supplementary Figs. 2, 3a). We also observe that the attractive potential energy term for nonpolar residues contributes similarly across each of the nine ex vivo fibril structures (Supplementary Fig. 5a; fa_atr). Lastly, we interpret the solvation energy term and find that on average alanine substitution of non-glycine and nonpolar residues yields favorable solvation energetics for all nine structures in both aggregate (Supplementary Fig. 4a) and individually (Supplementary Fig. 5b). This term indeed offsets some of the destabilizing contributions of mutating nonpolar residues to alanine but is relatively minor, suggesting that formation of voids and loss in the attractive potential energy dominates the energy change of nonpolar to alanine mutants. These data uncover the general folding and energetic principles of tau assemblies and relate these principles to the possible first steps in monomer folding, which may be defined by the burial of

nonpolar amino acids and the arrangement of lysine residues on the fibril surface.

### Identification of thermodynamic hotspots of aggregation

We next interpret energetics to understand on a per residue and motif level to gain insight into the rules that govern stability in fibril structures. To more easily visualize regions that may be important for stabilizing fibrils we calculated an average change in energy ($\Delta REU^{assembly}_{mut-wt}$) per residue for a five-residue window. We relate these values to amyloid propensity and find that many of the interactions between energetically important elements involve amyloidogenic segments including [275]VQIINK[280], [306]VQIVYK[311], [337]VEVKSE[342], and others (Fig. 4a). Indeed, these interaction hotspots also correlate to fragments that encode clusters of nonpolar residues (Fig. 4b). We also map the per residue energy change of mutation ($\Delta REU^{assembly}_{mut-wt}$) onto the nine distinct ex vivo fibril protofilaments and unsurprisingly observe that

residues that contribute to the largest change in energy are often buried in the fibril core and that residues on the periphery tend to have smaller energetic contributions (Fig. 4c). Consistent with this observation, we find alanine mutations significantly destabilize the attractive potential energy, indicating that burial of nonpolar residues in tau is a dominant contributor to fibril stability (Supplementary Fig. 4a; fa_atr). Additionally, we find that the solvation energy term generally decreases (i.e., becomes more favorable) with alanine substitutions because this allows favorable interactions with solvent, but these gains in solvation energy are often balanced with loss in vdW contacts. We fit the relationship between per residue change in stability to $\Delta SASA_{single\ layer}^{folding}$ and $\Delta SASA_{fibril}^{folding}$ as a function of amino acid types. We see that as expected, all amino acid types experience some degree of change in SASA upon folding into monomer conformation (i.e., as a single layer of the fibril), and then a further second stage of burial when that folded monomer conformation is incorporated as a layer of a fibril. Overall, the two terms cluster residues together, but the coefficients of determination ($R^2$) are low, suggesting that each residue type has a significant number of outliers that cannot be explained by the fit. For all residue types, there are some residues with high $\Delta SASA_{single\ layer}^{folding}$ and $\Delta SASA_{fibril}^{folding}$, but low impact on fibril stability as measured by $\Delta REU$ when mutated to alanine. This may be due to inwards-facing residues in the fibril that face either true voids or voids caused by unresolved densities, such as those in the CBD structures. We find that for both $\Delta SASA_{single\ layer}^{folding}$ and $\Delta SASA_{fibril}^{folding}$, the nonpolar residues have the largest baseline contribution in a change in energy per change in SASA, indicating that their burial is a major factor associated with fibril stability (Supplementary Fig. 5c). Both acidic and basic residues have populations with low change in energy when mutated to alanine but there is a subpopulation with both a large $\Delta SASA^{folding}$ and large change in energy. This indicates that there exists a set of buried charged residues that are important contributors to fibril stability. Indeed, there are several buried salt bridges that involve lysines and acidic residues that account for this behavior. For some fibril structures, including Huntingtin or CPEB protein, a large portion of buried contacts involves polar amino acids[67,68]. In our tau fibril analysis, it appears that nonpolar residues are largely buried in the structures leaving mostly polar and specifically basic residues on the surface with some exceptions (Supplementary Figs. 2c and 3a). Indeed, the unique protofilaments which have been identified across diseases leverage these surface residues to adopt alternate protofilament arrangements stabilized by the weak polar interactions that extend along the fibril axis (e.g., the Alzheimer's disease PHF/SF or the GGT type 1/2/3 fibrils).

To understand the energetics of the [306]VQIVYK[311] motif specifically, we looked at the energetics of the residues identified to interact with the amyloid motif in the different structures (Fig. 1). We find that in the PiD, CBD, and AGD structures, V306, I308, and Y310 from [306]VQIVYK[311] make stabilizing interactions with V337 and V339 from [337]VEVKSE[342] (Fig. 4d). Similarly, we find that in the AD-PHF, AD-SF, and CTE structures, V306, I308, and Y311 of the [306]VQIVYK[311] contribute stabilizing interactions with L376 and F378 of [373]THKLTFR[379] (Fig. 4e). In the PSP and GPT structures, Q307, V309, K311 of VQIVYK form stabilizing interactions with V350, S352, and I354 of [349]RVQSKIGS[355] (Fig. 4f). Additionally, we find that in CBD, AGD (and GGT) there are hotspot interactions between [306]VQIVYK[311] and [295]DNIKHV[300] mediated by nonpolar contacts centered on V306 and I297 (Supplementary Fig. 6a). Excitingly, this element was consistently formed in a core of fibril structures assembled from a recombinant fragment and is similar to the core element in the GGT structure[52]. These data suggest that [306]VQIVYK[311] plays important, stabilizing roles in diverse structural polymorphs observed in different diseases. The VQIVYK-based networks of interactions may help determine the fold and govern whether the fibril adopts local interactions with the [295]DNIKHV[300] sequence in two different ways (CBD, AGD vs. PSP, GPT), but also help establish long-range contacts to [337]VEVKSE[342] (CBD, AGD, and PiD) or

[373]THKLTF[378] (AD-PHF, AD-SF, and CTE). While [306]VQIVYK[311] seems to play a central role in all ex vivo (and nearly all recombinant with the exception of RNA-induced tau fibrils) fibrils, we also observe features that utilize other elements such as conserved and energetically stabilizing interactions between [355]DNITHV[363] and [370]KKIETH[376] in the GGT, PSP and PiD structures (Supplementary Fig. 6b)[69]. These interactions are peripheral to the central cores but are related to the [295]DNIKHV[300] and [306]VQIVYK[311] interaction because they similarly surround a β-turn capable PGGG motif. Understanding how these contacts are formed may help us facilitate controlling the order of interactions that lead to a fold observed in the disease.

To test whether the aggregation-prone [306]VQIVYK[311] peptide alone can engage in heterotypic interactions with other nonpolar elements predicted from the hotspot calculations, we designed peptide co-aggregation experiments using [306]VQIVYK[311] with [337]VEVKSE[342] and [350]VQSKIG[355], two different VQIVYK-interacting elements predicted to stabilize the different fibril structures. We used a ThT fluorescence aggregation assay and then compared endpoint fluorescence signals to understand whether any of these inert peptides can regulate the assembly of [306]VQIVYK[311]. We performed aggregation of the [306]VQIVYK[311] peptide at 200 μM alone, and in the presence of each heterotypic contact peptide at 200, 100, and 50 μM. We also measured the ThT signal of the test peptides at the three concentrations and confirmed the presence or absence of fibrils in all samples by TEM. (Supplementary Fig. 6c–f). We find that each peptide influences [306]VQIVYK[311] aggregation in different ways. The most dramatic effect was observed with [350]VQSKIG[355] (Fig. 4f; PSP and GPT) which blocked [306]VQIVYK[311] aggregation completely (Supplementary Fig. 6c). The [337]VEVKSE[342] peptide (Fig. 4e; CBD, PiD, and AGD) only inhibited VQIVYK aggregation at the highest concentration (200 μM) (Supplementary Fig. 6d). To further probe interactions between [306]VQIVYK[311] and [337]VEVKSE[342], we also co-aggregated [306]VQIVYK[311] with [337]VEVKSE[342] and two mutants of [337]VEVKSE[342] at hotspot positions mutating each valine to alanine. We also included an experiment with a soluble control peptide (sequence GSPSGS) with a proline intended to reduce β-sheet propensity. We find that [337]VEVKSE[342] yielded a similar decrease to that seen in the previous experiment, while the control peptide had no effect (Supplementary Fig. 6e). Interestingly, the AEVKSE peptide was not able to efficiently inhibit [306]VQIVYK[311] aggregation while VEAKSE was even more efficient (Supplementary Fig. 6e). These data suggest that regulating amyloid motif aggregation may be possible by leveraging peptide aggregation systems.

The intrinsically disordered tau protein has been shown to adopt discrete fibril conformations in disease[15,17,18,20–22]. Furthermore, these pathogenic conformations can be faithfully propagated in a prion-like manner from cell to cell, where in human disease the spread of tau pathology determines the progression of disease[10,70]. This process of transcellular spread can be mimicked in cellular systems by the transduction of extracellular seeds into cells to convert the intracellular naïve population of tau into pathogenic conformations of the same conformation as the source. Experimentally, this is performed by expression of wild-type (WT) tau as a fusion to a FRET-compatible fluorescent reporter (e.g., CFP/YFP or mClover/mCerulean3) in mammalian cell lines that can be used to induce aggregation of the intracellular reporter by transfection of a recombinant or a tauopathy patient-derived seed[71]. The extent of aggregation of the tauCFP and tauYFP inside cells can be quantified using FRET[71]. More recently, this system has been shown to replicate the conformation of tau seeds from cells into animal models and back into cells[10]. Leveraging the capacity of tau to propagate seed conformations in cellular systems, we wanted to test whether residues identified in our computational predictions are important for the formation of CBD tau fibril conformations in cells. We developed a system in which tau fused to CFP can be converted into a CBD-derived conformation by transduction of CBD human brain material. We hypothesize that tau mutations that are

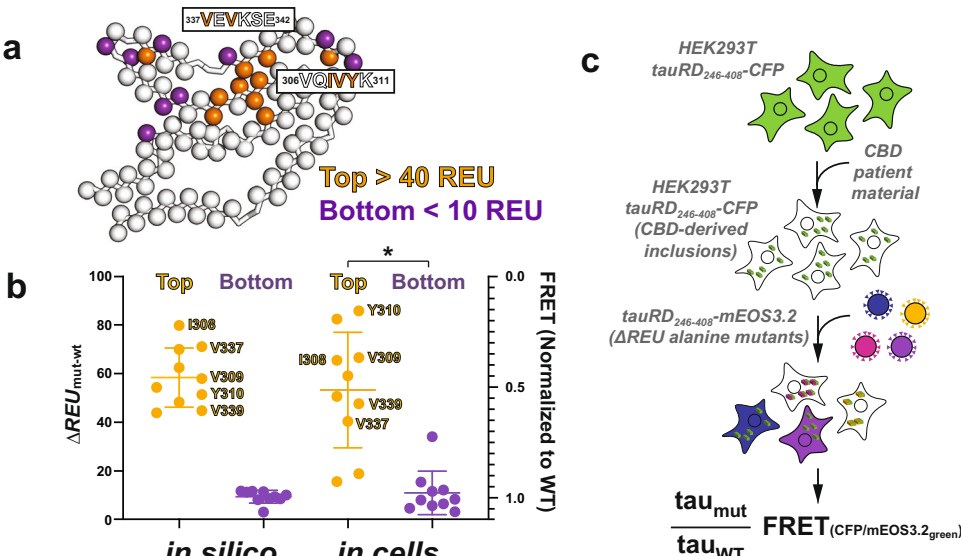

**Fig. 5 | Experimental validation of predicted CBD hotspots. a** Cartoon schematic to evaluate the importance of key amino acids in cellular tau seeding experiments using CBD patient tissues. Orange residues are predicted to be important to fibril stability (normalized $\Delta REU_{mut-wt}^{assembly}$ >40), while purple residues are predicted to be minimally important to fibril stability (normalized $\Delta REU_{mut-wt}^{assembly}$ <10). **b** Plot comparing the top-10 and bottom-10 most important residues as determined by $\Delta REU_{mut-wt}^{assembly}$ from the in silico alanine scan (scaled from normalized 0 $\Delta REU_{mut-wt}^{assembly}$, i.e., no predicted loss in fibril stability caused by alanine mutant to normalized 100 $\Delta REU_{mut-wt}^{assembly}$ –high predicted loss in fibril stability due to alanine mutant), alongside the effect of these alanine mutants on the ability of CBD-brain derived tau seeds to seed aggregation as measured by the biosensor-based aggregation assay.

(Range from 1.0 = no change in FRET, i.e., no effect on the ability of mutant tau to aggregate in biosensor cells when seeded with CBD-derived seeds as compared to the WT, to 0.0 = no FRET observed, i.e., mutant tau is unable to induce FRET-detectable aggregation in the biosensor). The experiment was performed in triplicate and the data are shown as averages with standard deviations. The experimental top and bottom hits are shown to be significantly different with $p$-value 0.0016 ($U$ = 2.0) using a two-tailed Mann–Whitney test and are indicated by a star. The fibril incorporation assay was performed once. **c** Cartoon schematic showing the method of the in vitro, FRET biosensor-based tau seeding cell assay used to assess the impact of mutants on tau's ability to aggregate when seeded by proteopathic tau seeds.

important for the incorporation of tau into the CBD-derived aggregates would prevent incorporation and therefore not yield FRET while mutations that are not important would yield FRET signatures similar to WT tau. We used this system to test our top 10 $\Delta REU_{mut-wt}^{assembly}$ predicted hits (Fig. 5a, b, orange) identified in the CBD in silico alanine scan and as a control, we also tested 10 neutral $\Delta REU_{mut-wt}^{assembly}$ hits (Fig. 5a, b, purple). Consistent with our observations, the top hits from our calculations cluster to the contacts between the $^{306}$VQIVYK$^{311}$ and $^{337}$VEVKSE$^{342}$ motifs (Fig. 5a). We produced cell lines expressing WT tau fused to CFP which were treated with CBD patient material to replicate the CBD conformation in the cells. Subsequently, we transduce tau fused to mEOS3.2[72] encoding alanine mutations to test the ability of each mutation to incorporate into the WT-propagated CBD-derived inclusions in the cells. Mutations that interfere with incorporation into CBD-derived inclusions should not yield FRET between CFP and mEOS3.2 while mutations that can incorporate into inclusions should have similar FRET to WT tau. To estimate the effect of these mutations, we compare the FRET signal of the mutant to the WT tau (Fig. 5c and Supplementary Fig. 4f). We find that the experimental data compare well with the $\Delta REU_{mut-wt}^{assembly}$ calculations and similarly allow separation of the top hits from the bottom hits with significant $p$-values (Fig. 5b). For the top hits we predict correctly 8 of 10 positions which cluster to the important amyloid motif interactions (Fig. 5b). The 2 predicted positions at F346A and P312A that do not match the experiment, fall outside of the core of interactions, highlighting the potential failure of the energy function to capture the energetics of these contacts correctly or inability of our method to model larger conformational shifts that may be caused by mutation. Our experiments leverage tau's capacity to replicate fibril conformations in cells and allow us to experimentally validate our computational method to discover hotspot residues important for fibril stability.

Through computational alanine scanning, we can identify hotspot residues and sequence regions that form the heterotypic interactions that stabilize fibril structures. We observe these interactions in in vitro aggregation experiments and validate the importance of hotspot residues to fibril formation using an in-cell model of tau amyloid aggregation. We anticipate that stabilizing these interactions within a unique tau fold while destabilizing them in other structures will restrict the tau monomer to adopt only that single tau fibril fold.

## Classification of structures and their features based on their energetic profiles

We next employed a machine learning approach to begin classifying the structures based on the per residue $\Delta REU_{mut-wt}^{assembly}$ values to learn what residues/features may help cluster and discriminate fibril conformations. As some of the structures differ in sequence composition, we first curated the $\Delta REU_{mut-wt}^{assembly}$ data to only contain residues present in all of the structures. For example, PiD fibrils are comprised of only a 3R tau isoform lacking the second repeat domain, AD/CTE are limited to repeat domains three and four, while others (AGD, CBD, GGT, PSP, and GPT) contain sequences from repeat domain two, three, and four. From this reduced set of common residues, we calculated a distance matrix of all 14 disease-derived tau fibril polymorphs, which include subtypes for some of the structures including AD, AGD, GGT, and GPT (Supplementary Fig. 7a). This distance matrix was then used to construct a hierarchical clustering of the different fibril types (Fig. 6a and Supplementary Fig. 7b). We find that related subtypes cluster together and reassuringly structures such as AD-PHF, AD-SF and CTE or AGD_T1, AGD_T2, and CBD cluster together. Likewise, the GPT and GGT structures cluster together with PSP. Again, this observation is consistent with the GPT structure being described as a hybrid PSP and GGT structure. Consistent with our prior observations these more related structures also contain conserved interactions involving $^{306}$VQIVYK$^{311}$. Interestingly, the PiD structure clusters together

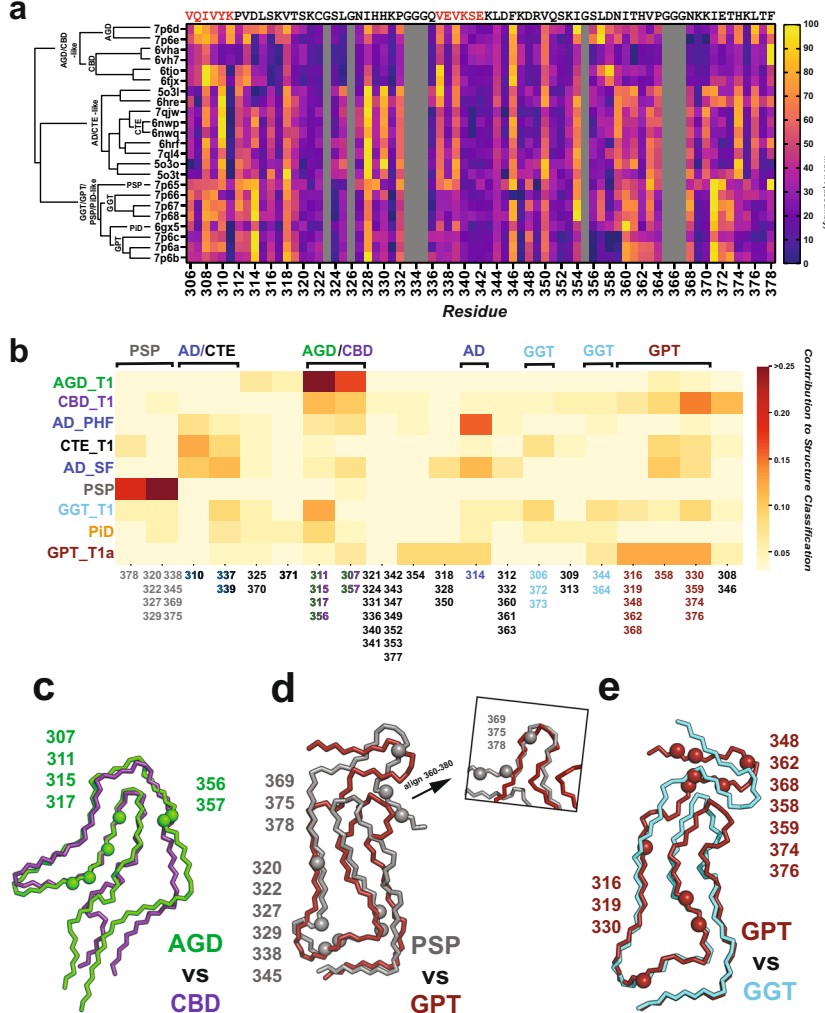

**Fig. 6 | Feature classification clusters structural polymorphs by hotspot interactions and folds. a** Dendrogram produced by hierarchical/agglomerative clustering using Ward's method of tau fibril structures based on normalized in silico $\Delta REU_{mut-wt}^{assembly}$ values determined by the alanine scan, displayed next to the plot of normalized $\Delta REU_{mut-wt}^{assembly}$ values. Scale is colored in plasma from yellow (highest in silico $\Delta REU_{mut-wt}^{assembly}$) to purple (lowest in silico $\Delta REU_{mut-wt}^{assembly}$) for all available tau ex vivo fibril structures from AD, CBD, AGD, PSP, GPT, GGT, and PiD, including subtypes. PDB id: 7p6d, 7p6e, 6vha, 6vh7, 6tjo, 6tjx, 5o3l, 5o3t, 6hre, 7qjw, 6nwp, 6nwq, 6hrf, 7ql4, 5o3o, 5o3t, 7p65, 7p66, 7p67, 7p68, 6gx5, 7p6c, 7p6a, and 7p6b. **b** Contribution of residues/groups of residues towards the correct classification of an aggregate structure by a random forest classification model trained on the in silico $\Delta REU_{mut-wt}^{assembly}$ of alanine mutants. Clusters most important for the correct identification of a structure are colored red, while the least important clusters for a structure's identification are colored pale yellow. Residue labels are colored by the structure type they most contribute to the identification of- CBD (purple), AGD (green), AD PHF/SF (blue), CTE (black), PSP (gray), GGT (cyan), PiD (orange), or GPT (red). Less important residue clusters for classification/identification are colored light gray. **c**–**e** Overlaid fibril monomers of similar structure with residues that most contribute to their classification are shown with C-α atom spheres. **c** Overlay of CBD (blue) with AGD (green), highlighting CBD residue 358 with C-α atom shown as a blue sphere and AGD residues 311, 315, 317, 356 C-α atoms shown as green spheres. **d** Overlay of PSP (gray) with GPT (red), highlighting PSP residues 307, 320, 322, 327, 329, 338, 351, 375, 378 with C-α atoms shown as gray spheres and GPT residues 319, 348, 354, 359, 362, and 376 C-α atoms shown as red spheres. **e** Overlay GPT (red) with GGT (cyan), highlighting GPT residues 319, 348, 354, 359, 362, and 376 C-α atoms shown as red spheres and GGT residues 354 and 357 C-α atoms shown as cyan spheres.

with GPT. This anomalous clustering of PiD is perhaps not surprising because of this conformation's unique sequence composition, involving residues from the first repeat domain. As such, a portion of the sequence is unique and cannot be used in clustering and structure classification with this method. After the identification of common features, we focused our efforts to tease out more subtle differences between related structures which would be difficult to discriminate via the surface using proteins (i.e., antibodies). Being able to predict residues that form energetically discerning interactions may allow the design of tau sequences at specific sites to stabilize one conformation while destabilizing another, allowing the creation of designer tau sequences that can only propagate a single conformation while being incompatible with others.

## Identification of differentiating features among disease-associated tau fibrils

We proceeded to the identification of distinguishing features between tau fibrils by training and interpreting a machine learning model on the per-residue $\Delta REU_{mut-wt}^{assembly}$ values generated by the in silico fibril alanine scan. Briefly, we performed hierarchical clustering on the residue $\triangle REU_{mut-wt}^{assembly}$ values and noted that many sets of residues had highly similar behavior between fibrils (Supplementary Fig. 7b). To reduce data dimensionality and simplify the interpretation of the classification model by leveraging this covariance of residues, we proceeded to perform feature agglomeration, recursively combining covarying groups of residues into single features (Supplementary Fig. 7c). With this reduced set of features, we trained a random forest classification model

model to classify the various disease-associated fibrils and subtypes into their parent fibril types (e.g., GGT_T1/T2/T3 as GGT, AD-PHF/SF as AD, GPT_T1a/T1b/T2 as GPT, etc.). 2500 classification models were generated using 25% of the replicates generated by the alanine scanning protocol as training data and the remaining 75% of the replicates as testing data to verify the performance of the classification model, selecting the model with the highest classification accuracy on the testing set. A model was selected with >99.5% prediction accuracy on the test set, showing the ability to effectively discriminate structures based solely on their $\Delta REU_{mut-wt}^{assembly}$ values. Despite its highly accurate classification performance, the model is least confident in its classification of AD-PHF/AD-SF vs. CTE_T1 fibrils (Supplementary Fig. 7d). This is likely due to the high degree of mutual similarity of the AD-PHF, AD-SF, and CTE_T1 structures (and thus the $\Delta REU_{mut-wt}^{assembly}$ values derived from them).

Subsequently, this model was used to identify major distinguishing features of each disease fibril type by running the classifier model on the per-residue $\Delta REU_{mut-wt}^{assembly}$ values (mean of all 35 replicates) and inspecting the model to see to which degree each residue cluster contributed to the correct classification of each structure (all structures were correctly classified by the model) (Fig. 6b). Notably, this reveals clusters/sets of residues that have unique behavior in certain fibrils/sets of fibrils as discerned by the model and used in the identification of the structure. We find that residues Q307, K311, L315, K317, S356, and L357 are strongly distinctive in AGD/CBD. We see a weaker contribution of single clusters of residues to the prediction of AD vs. CTE fibrils, again suggesting that differences between the two are relatively minor and forcing the model to parse out smaller differences over a larger number of fibrillar features to make an accurate classification. We map these values onto the structures and note that these discriminative positions highlight locations where fibril structures deviate from each other. For example, AGD-discriminating residues lie where the residues [312]PVDLSK[317] interact [285]SNVQSKGS[293], a region with differing backbone geometry between AGD and CBD. Similarly, residues S356 and L357 are also located in a region that deviates between AGD and CBD (Fig. 6c). Taking the AD/CTE fibrils together, differentiating residues here highlights the residues differing in the one-vs.-two-sided [306]VQIVYK[311] interaction that separates AD, CTE, and PiD from the other fibrils. These discriminative residues also localize to regions of diverging structure between PSP/GPT and GGT/GPT, suggesting the classification model is successfully identifying regions of distinction between tau fibril structures solely based on the per-residue $\Delta REU_{mut-wt}^{assembly}$ values. We anticipate that accurate identification of distinguishing regions of tau fibrils can contribute to the design and development of cellular tau-based reagents that encode sequences that distinguish fibril conformations and more favorably adopt desired fibril conformations, as well as helping lend insight into the differing disease processes that produce these unique conformers.

## Discussion

We have developed an in silico alanine scanning method to interpret the energetic contribution of amino acids in fibrillar structures obtained using cryo-EM methods. Our work presents a comprehensive analysis of cryo-EM tau fibril structures that uncovers amino acids (i.e., hotspot residues) that stabilize intra-peptide interactions in the distinct structural polymorphs. These hotspot residues are often non-polar and lie in sequences that are either amyloidogenic or interface with amyloidogenic motifs. Interestingly, we discover that these residues form interaction motifs that are often modular, being combined in various ways across the different fibril structures (Fig. 7). Related fibril conformations use similar sets of these hotspot interactions. This modularity of interactions uncovers the underlying interactions that enable the different fibrillar folds in various structural polymorphs. We use cellular and in vitro aggregation experiments to validate energetically important interactions predicted by our methods. We also used

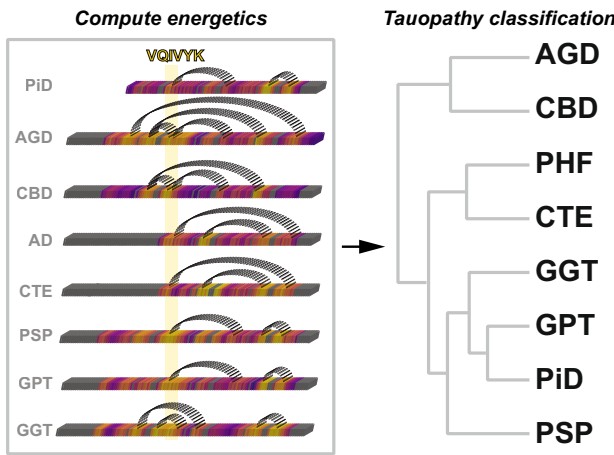

**Fig. 7 | Classification of tau fibril structures using per residue energetics.** Contacts observed in fibrils between energetically important hetero-typic interactions determine tauopathy folds. The linear heatmaps are shown as 3D rendered boxes that are colored by the normalized change in energy values (yellow = stabilizing and purple = neutral). Dashed semi-circles link contacts between hotspot regions that interact in the different structures and are used to classify the different folds.

the per residue energy contributions to classify diverse tau fibril structures and uncover sets of modular interaction motifs that are key features that permit the clustering of related structures and conversely differentiate distinct folds. These uncovered interactions mirror our initial observation surrounding the [306]VQIVYK[311] motif. We anticipate that our method will be broadly useful for the interpretation of folding and interaction energies in fibril structures and acknowledge that future work must focus on developing experimental methods to capture these energetics—either directly or by proxy—to improve our predictions. These data will be important to interpret how pathogenic mutations alter the conformation of proteins and set the stage to begin designing protein sequences that preferentially adopt unique structural folds.

Evidence is emerging that distinct tau fibril structural polymorphs are associated with different diseases[15,17,18,20-22]. This link between conformation and disease is now unambiguous but many questions remain. It is unknown how these fibril structures form in disease, what is the relationship between structure and the capacity to spread prion-like manner, and how these distinct fibril conformations promote region-specific neurodegeneration that ultimately links the pathology to the clinical presentation of the disease. Recombinant reproduction of disease fibril conformations remains a challenge, highlighted by the stark differences between ex vivo patient conformations compared to the heparin-derived recombinant forms[16]. Recent work on recombinant tau fragments showed some promise in recapitulating AD and CTE tau fibril conformations but this must be understood both in the context of full-length proteins and the intracellular environment[52]. Thus, understanding how these structures are formed remains at the forefront of identifying disease mechanisms. Knowledge of these structures may be used to detect tauopathy-specific fibril conformations to diagnose disease and treat tauopathies in a conformation-specific manner. For example, antibodies can be selected that have specificity towards structural polymorphs, or alternatively cellular systems of tau propagation can be engineered to selectively detect different fibril folds. Furthermore, knowledge of the misfolding pathways towards these conformations will likely provide additional insight into mechanisms of prion-like propagation of each tau conformation within a cell and how each fibril conformation may interact with surface receptors to promote transcellular propagation[73]. Tau fibrils are thought to interact with heparan sulfate proteoglycans on cell surface receptors[74,75]. Recent data have suggested that the AD fibril

conformation may be spread by interacting with the LRP1 receptor[73], but it is not known whether this mechanism is relevant to other tauopathy fibril conformations. Indeed, these insights may reconcile how different fibril structures may target different cell types leading to the characteristic pathology and pattern of neurodegeneration that ultimately defines each clinical syndrome.

Our analysis uncovers fundamental rules that underlie the modularity of contacts in the fibril structures centered on the $^{306}$VQIVYK$^{311}$ motif, which functions as a hub of interactions. Our data support that the central role of $^{306}$VQIVYK$^{311}$ is defined by the combination of both its amino acid composition and the ability to adapt β-sheet conformations in solution, allowing other nonpolar residues to engage in contacts with specified geometries on two surfaces. Work from van der Kant et al. showed that amyloid motifs are important for many other fibril structures, underscoring the central roles of amyloid motifs in stabilizing fibril conformations in tau and other fibrils[42,43]. In that study, the authors also identified the $^{306}$VQIVYK$^{311}$ motif as energetically important using orthogonal force fields, highlighting the generality of our findings[43]. We also demonstrated $^{306}$VQIVYK$^{311}$ interacting motifs can modify VQIVYK assembly in vitro and mutations that disrupt these networks of interactions block the incorporation of monomers into assemblies. Interestingly, a corollary derived from our work suggests that monomers may form intramolecular interactions compatible with a fibril fold. Tau's capacity to sample these fibrillar-like conformations in solution suggests that it may be possible to reverse engineer the tau sequence to stabilize substructures that recapitulate those conformations in ex vivo fibrils, thus controlling the formation of different folding steps on the path to fibrillar folds. Knowledge of the misfolding tau intermediates that are on the pathway to disease-specific fibril formation may be useful to identify species to target for diagnosis and disease treatment. Recent successful efforts to fibrillize a recombinant fragment containing the third and fourth repeat domains of tau into AD and CTE disease conformations highlight how limiting the number of interactions through the usage of fragments may regulate tau folding[52]. Inclusion of specific anions/cations in the reaction mixture may allow further reduction of possible conformations by a combination of charge neutralization and stabilization of nonpolar contacts. However, more complex topologies with longer sequences (and therefore more possibilities for the formation of different polymorphs) will likely remain a challenge to recapitulate in vitro[52]. It is possible to propagate disease-derived tau fibril conformations in cells, making it possible to screen for cellular factors, such as proteins, ligands, or post-translational modifications (i.e., phosphorylation, acetylation, ubiquitination, etc.) involved in the maintenance of tau fibril conformations in different cell types known to harbor tau inclusions in tauopathies. Conceptually these interactions may simply stabilize pro-aggregation conformations to yield tau conformations that are on a pathway toward distinct fibril folds. Understanding the network of interactions responsible for the stability of fibrils may also provide insight into the physiological states and structural intermediates necessary to stabilize certain interactions and preferentially form specific fibril polymorphs.

The mechanism that underlies tau's capacity to adopt a diversity of structures is balanced by intramolecular interactions that limit its ability to assemble. The full-length tau protein is remarkably aggregation-resistant, yet in solution, it must transiently sample conformations compatible with fibrillar-compatible conformations. It is thus tempting to speculate that the formation of disease-associated fibrils is simply a result of the stabilization of those pro-aggregation states that overcome the protective conformations. Tau's disordered structural ensemble must populate both pro- and anti-aggregation states and deeper knowledge of fibril structures and the interactions that stabilize them may help parse the features required to differentiate these populations. Our prior work implicated that tau monomers can adopt stable conformations that are aggregation-prone. A proposed hallmark of this pathogenic state is the exposure of the fibril-forming $^{306}$VQIVYK$^{311}$ motif[41]. Thus, exposure to the $^{306}$VQIVYK$^{311}$ may be the defining feature of pro-aggregation species in early disease[40]. Deposition of tau amyloids in frontotemporal dementia-tau (FTD-tau) is linked to mutations in the microtubule-associated protein tau gene (MAPT) that often localize to the repeat domain and in proximity to amyloid motifs[70,76]. We have proposed a mechanism in which pathogenic tau mutations promote exposure of the amyloid motifs—including $^{306}$VQIVYK$^{311}$—by destabilizing a beta-turn motif and thereby promoting rapid tau aggregation. This has recently been supported by observations derived from an experimentally derived tau ensemble comparing wild-type and mutant tau[77]. Therefore, knowledge about how the tau monomer ensemble changes in response to mutations may provide mechanistic insight into the initial conversion of tau into pro-aggregation states. While to date there are no tau fibril structures encoding FTD-tau mutations, we propose that the mutations perturb monomers towards states compatible with the fibril state. We predict that deeper knowledge of the fibril conformation will inform the required interactions that a monomer must adapt to incorporate into a fibril.

We compare the current explosion of cryo-EM fibril structures to the early efforts of X-ray crystallography to determine the structures of globular proteins. These initial structural efforts created the structural knowledge that is now used to develop improved algorithms to accurately predict protein structure from sequence and to design de novo protein structures with novel functions. Combined with evolutionary sequence information, we anticipate that as the cryo-EM fields focus on the structure determination of fibrils matures, new computational methods will be developed that will accurately predict how sequences can convert from disordered ensembles of conformations to possible fibrillar conformations. Further advances in structure determination and protein modeling will begin to explore how one sequence can adopt an ensemble of different conformations and the requirements to understand rate-limiting interactions that determine the formation of structural folds. Additionally, many current amyloid fibrils contain unresolved electron densities, and advances in the identification and/or modeling of these features will further improve understanding of fibril formation and structure. These concepts can be leveraged using computational protein design to generate protein sequences that can bias these ensembles towards a more defined state compatible with a single fibrillar conformation and not compatible with other conformations. Applications of these designed "restrictive" sequences may allow selective amplification of these structures, using cellular systems of tau and other protein prion-like propagation[71,78–80] offering avenues to diagnose disease based on conformation in pre-mortem samples. These ideas may further inform the capacity of seed to propagate to promote pathology thus relating macroscopic properties such as the shape of protein inclusions in human tissues to molecular properties of the fibrillar structure.

Together our experiments and computational method have uncovered how energetically important interactions that recur across different structural polymorphs may contribute to the folding of tau into distinct structural polymorphs. These methods will help uncover the folding mechanisms of tau and other proteins into conformations associated with the disease. Our energy calculations can be used to pinpoint energetically distinguishing interactions thus allowing the design of sequences that can propagate only one fibril conformation

that may be useful for the development of diagnostic reagents. Finally, our computational method is generalizable to any fibrillar structure and could be used to uncover energetically important contacts to begin classifying structures by their stabilizing interactions, as well as identifying more general patterns in amyloid aggregation and fibril structure within the entire amyloid proteome.

## Methods

All of the research performed in this study complies with ethical regulations. All human samples utilized in these experiments were derived from deceased subjects. The UT Southwestern Institutional Review Board has determined that such studies are exempt from human subjects regulations as codified in federal law. All autopsies were performed only with permission of the decedent's next of kin or other person(s) legally authorized to provide such permission. All human materials utilized in these experiments were deidentified and no individually identifiable protected health information was available to the investigators using these materials.

### Amyloid motif prediction in tau and in vitro ThT peptide fibrillization

We used the ZipperDB database to identify possibly amyloidogenic motifs in tau[48]. Hexapeptides that showed REU below −25 were selected for subsequent in vitro aggregation experiments. Energies for fragments that contain prolines are omitted in the calculations as they lead to large VdW energetic penalties. Additionally, we used Pasta[50] and Waltz[47] to determine consensus amyloid motifs in tau. C-terminal amidated and N-terminal acetylated hexapeptides representing regions of amyloidogenic behavior (as predicted by ZipperDB) were sourced from GenScript at ≥95% purity. The peptides were disaggregated in 300 μL of trifluoroacetic acid (TFA) and incubated at 30 °C with 850 RPM orbital shaking for one hour. After disaggregation, peptides were blown dry of TFA under $N_2$ gas and then lyophilized for 2 h to remove residual TFA. For fibrillization, lyophilized peptides were resuspended in 500 μL of ultrapure water, vortexed thoroughly to resuspend, and then diluted (based on initial mass) to a final concentration of 1 mM peptide in 1X phosphate-buffered saline (PBS) using 50 μL of 10x PBS + NaOH (1.37 M NaCl, 27 mM KCl, 100 mM $Na_2HPO_4$, 18 mM $KH_2PO_4$, 1.56 μM NaOH) and ultrapure water. pH was verified to be neutral using pH test strips. Fibrilization was carried out for one week at 37 °C with 900 RPM orbital shaking. Samples were taken from the in vitro fibrilization reactions for ThT measurement in 6-plicate. In the dark, 3 μL of 250 μM Thioflavin T (ThT) solution was mixed with 27 μL of the in vitro fibrilization reaction mix for a final ThT concentration of 15 μM. Six blank samples containing only PBS with 15 μM ThT were also prepared. The samples were loaded onto a clear-bottom, 384-well plate and read in a Tecan Infinite M100 at 446 nm excitation wavelength (5 nm bandwidth), 482 nm emission wavelength (5 nm bandwidth). The instrument was heated to 37 °C, and samples were shaken for 10 s prior to the acquisition of the data. The mean blank sample fluorescence was subtracted from the fibrillized peptide fluorescence and reported as the ThT signals for the samples.

### In vitro ThT aggregation of VQIVYK alanine mutant and co-aggregation of VQIVYK with competing peptides

For heterotypic VQIVYK co-aggregation experiments and aggregation of VQIVYK alanine mutants, all peptides were monomerized in TFA (as above). Reactions containing sufficient peptide for 350 μL of 200 μM VQIVYK alone, 200 μM VQIVYK alanine mutants, 200 μM VQIVYK with 50 μM, 100 μM or 200 μM VEVKSE, VQSKIG, AEVKSE, VEAKSE, GSPSGS peptides, or 200 μM VEVKSE, VQSKIG AEVKSE, VEAKSE or GSPSGS peptides alone were mixed and then blown dry of TFA under $N_2$ gas and then lyophilized for 2 h to remove residual TFA. In the dark, a 1X

PBS + 25 μM ThT solution was prepared. Reactions were prepared by resuspending the mixed, lyophilized peptides in 350 μL 1X PBS + 25 μM ThT. The samples were loaded in six replicate, 50 μL reactions along with 6, equally sized blank reactions containing only 1X PBS + 25 μM ThT onto a clear-bottom, 384-well plate and read in a Tecan Infinite M100 at 446 nm excitation wavelength (5 nm bandwidth), 482 nm emission wavelength (5 nm bandwidth). The instrument was heated to 37 °C, and samples were shaken for 10 s prior to the acquisition of each data point. Data points were collected every 5 min for the first 8 h and then every 30 min afterward for one week and the blank fluorescence was subtracted. The endpoint ThT values was plotted with Graphpad Prism 9.4.1.

### Transmission electron microscopy

An aliquot of 5 μL sample was loaded onto a glow-discharged Formvar-coated 200-mesh copper grids for 30 s and was blotted by filter paper followed by washing the grid with 5 μL ddH₂O. After another 30 s, 2% uranyl acetate was loaded on the grids and blotted again. The grid was dried for 1 min and loaded into an FEI Tecnai G2 Spirit Biotwin TEM. All images were captured using a Gatan 2Kx2K multiport readout post-column CCD at the UT Southwestern EM Core Facility.

### In silico rosetta $\Delta\Delta G^{\text{interface}}$ and $\Delta\text{REU}^{\text{assembly}}_{\text{mut−wt}}$ calculation with backrub sampling

Varying numbers of layers of fibrils protofilaments were prepared for use in in silico estimation of assembly energies. A selection of tau fibril structures was retrieved from the Protein Data Bank (PDB) including PDB IDs: 7p6d, 7p6e, 6vha, 6vh7, 6tjo, 6tjx, 5o3l, 5o3t, 6hre, 7qjw, 6nwp, 6nwq, 6hrf, 7ql4, 5o3o, 5o3t, 7p65, 7p66, 7p67, 7p68, 6gx5, 7p6c, 7p6a, 7p6b, 6qjh, 6qjm, 6qjp, and 6qjq. For fibrils with two symmetric protofilaments, a single protofilament was selected for in-silico analysis to reduce the required computational resources. Using Pymol (version 2.4), fibril PDB structures were created with numbers of layers varying between three and nine using the above-mentioned PDB-deposited structures[81]. The deposited and symmetrized assemblies were used to generate assemblies ranging from three to nine layers. Briefly, we used structural alignment to superimpose the top two chains of the deposited fibril with the bottom two chains from a duplicated fibril assembly, preserving the geometry of the assembly while extending the fibril length. Overlapping chains were removed and chains were renamed to produce assemblies of the desired number of layers with chain lettering increasing from the top to the bottom layer. These assemblies were then used as input for the subsequent mutagenesis and minimization using the RosettaScripts interface to Rosetta v3.12[53].

Changes in assembly energy were calculated using a method adapted from the Flex ddG protocol described by Barlow et al. which was originally developed to interpret the stability of protein interfaces by comparing the energetics of unbound components and bound complex[59]. This approach employs a "backrub" minimization in the proximity of the mutation that allows additional optimization of the interactions that are performed on the mutant and WT structures. As the original approach computed differences between the unbound and bound states of a complex, we were able to adopt this approach to the calculation of interface energetics ($\Delta\Delta G^{\text{interface}}$) within fibrils as well as total energetics of the assemblies. First, a definition file was generated that describes the chains of the PDB to mutate to alanine and sets of chains that defined a subunit interface (used for $\Delta\Delta G^{\text{interface}}$ calculations). For both the interface and the $\Delta\text{REU}^{\text{assembly}}_{\text{mut−wt}}$ calculations, all chains were mutated. For the $\Delta\Delta G^{\text{interface}}$ calculations on nine-mer assemblies presented, the chains of the center three layers of the nine-mer were used to define the interface. From the input assembly, a set of pairwise atom constraints with a maximum distance of 9 Angstrom were generated with a weight of 1, using the fa_talaris2014 score

function. Using this constrained score function, the structure then underwent minimization. After minimization, the residues within 8 angstroms of the mutation site underwent backrub sampling to better capture backbone conformational variation. These sampled structures were either only repacked and minimized, or the alanine mutation was introduced, followed by repacking and minimization. The Rosetta InterfaceDdgMover was used as in the Flex ddG protocol to allow an analysis of the $\Delta\Delta G^{\text{interface}}$ by defining a fibril interface, giving the lowest energy bound and unbound states for both a wildtype and mutant fibril. For $\Delta\text{REU}_{\text{mut}-\text{wt}}^{\text{assembly}}$ calculations, the bound wild-type and bound mutant structures reported by the interface ddg mover were used for estimating the change in assembly energy due to an alanine substitution. This is repeated for 35 independent replicates. The lowest energy bound mutant and bound wild-type structure energies from each replicate were extracted, and the change in energy was calculated by subtracting the wild-type, non-mutagenized assemblies' energy from the mutant assemblies' energy. The mean change in energy over the 35 replicates was reported as that residue's $\Delta\text{REU}_{\text{mut}-\text{wt}}^{\text{assembly}}$. Additionally, we compared energetics of the internal $\Delta\text{REU}_{\text{mut}-\text{wt}}^{\text{internal}}$ and edge $\Delta\text{REU}_{\text{mut}-\text{wt}}^{\text{edge}}$ layers from the AD-PHF and CBD layer series ($n = 3$ to $n = 9$). Glycine residues were excluded from the analysis because mutation to alanine at these sites yielded clashes not resolvable with minimization and backrub sampling (Supplementary Fig. 3e–g). To calculate $\Delta\Delta G^{\text{interface}}$ for a residue, the Flex ddG protocol was used as described by Barlow et al.[59]. Briefly, the $\Delta G$ of binding was calculated by subtracting the bound state energy from the unbound state energy for both a wildtype and mutant fibril, and the $\Delta G_{\text{wt}}$ was subtracted from the $\Delta G_{\text{mut}}$ to yield the $\Delta\Delta G^{\text{interface}}$ for that alanine mutant.

$$\triangle\triangle G^{\text{interface}} = \triangle G_{\text{mut}} - \triangle G_{\text{wt}} = [\text{bound} - \text{unbound}]_{\text{mut}} - [\text{bound} - \text{unbound}]_{\text{wt}}$$

$$\Delta\text{REU}_{\text{mut}-\text{wt}}^{\text{assembly}} = \text{bound}_{\text{mut}} - \text{bound}_{\text{wt}}$$

The $\Delta\Delta G^{\text{interface}}$ and $\Delta\text{REU}_{\text{mut}-\text{wt}}^{\text{assembly}}$ for a given residue were then averaged over 35 replicates to yield the final values for the residue. This procedure is repeated for every residue in the structure to generate a set of $\Delta\Delta G^{\text{interface}}$ and $\Delta\text{REU}_{\text{mut}-\text{wt}}^{\text{assembly}}$ values for all residues in each fibril structure. For internal and edge layer analysis, the same procedure was performed, but excluding residues either on the edge chains or the center chains when calculating $\Delta\text{REU}_{\text{mut}-\text{wt}}$.

### Structure backbone RMSD analysis

Backbone RMSD analysis of the output structures was carried out by extracting structures used to calculate each replicate's change in assembly energy (the lowest energy bound mutant and bound wildtype structures for each replicate), using the Flex ddG extract_structures.py script[59]. The extracted structures then had their RMSDs calculated for the PDB chains of interest using the BioPython PDB module to create a list of the backbone atoms (c-α, amide nitrogen, carbonyl carbon, and carbonyl oxygen) for each residue in the chains of interest of both the input structure and the generated structures and using the BioPython PDB. Superimposer module to calculate the RMSD between the two sets of backbone atoms[82]. The resulting RMSDs were then saved by residue for each structure for all 35 replicates, allowing by residue and by structure visualization of the structural deformation allowed by the alanine-scanning protocol.

### Per-residue ΔSASA calculations

Per-residue ΔSASA calculation was performed by utilizing three states for each fibril- the extended monomer (a linear peptide fragment of the fibril generated by using the Pymol 'fab' command with the fibril sequence), a monomer in the fibril conformation extracted from the CryoEM fibril structure, and a trimer of the fibril. The SASA of each state was calculated in Rosetta, with the RosettaScripts 'TotalSasa' mover with the 'report_per_residue_sasa' flag set to true[53]. This generated a per-residue SASA for each residue of the extended monomer, fibril conformation monomer, and all the residues in the fibril multimer. For the fibril multimer, the SASA of the residues in the central chains was extracted. Then, the per-residue $\triangle\text{SASA}_{\text{folding}}^{\text{monomer}}$ was calculated by subtracting each residue's SASA in the folded monomer from the SASA in the extended monomer, and the $\triangle\text{SASA}_{\text{folding}}^{\text{in fibril}}$ was calculated by subtracting each residue's SASA in the folded monomer in the center of the fibril from the SASA in the extended monomer.

### Preparation of lysates from CBD patient material

Ten percent weight by volume brain homogenates were made from the frontal cortex of CBD cases using Power Gen 125 tissue homogenizer (Fischer Scientific) in 1XTBS buffer supplemented with cOmplete ULTRA protease inhibitor cocktail tablets, EDTA free (Roche). The lysates were sonicated in a water bath (QSonica) for 10 min, for 1 min "on" and 30 s "off" intervals, at an amplitude of 65, and centrifuged at $21,000 \times g$ for 15 min at 4 °C to remove the debris. The supernatant protein concentration was quantified using a Pierce 660 nm Protein Assay Reagent (Thermo Scientific), and subsequently used in seeding assays (see below).

### Construction of WT tauRD (246–408) cell line, seed amplification, and incorporation assay

Lentivirus containing the human WT tauRD (residues 246–408), C-terminally fused to a cyan fluorescent protein (CFP), was made in order to constitutively express the WT tauRD-CFP protein in HEK 293-T cells (ATCC CRL-1268). Lentivirus was made as follows: 7.5 μL of TransIT (Mirus) was mixed with 142.5 μL of Opti-MEM (GIBCO) for 5 min.This mixture was combined with psPAX2 (1200 ng), VSV-G (400 ng), and FM5-CMV-WT tauRD-CFP (400ng) plasmids and incubated for another 25 min before addition into a six-well dish containing HEK 293 T cells that were plated with 300,000 cells, 24 h prior. Cell media (10% FBS, 1% Pen/Strep, 1% GlutaMax in Dulbecco's modified Eagle's medium) containing virus was collected after 48 h, centrifuged at $100 \times g$ for 5 min, and supernatants were aliquoted before freezing at −80 °C. HEK 293-T cells were then transduced with the lentivirus and single cells were sorted into 96-well plates after the expression of the protein was confirmed. The cell line was expanded and used for downstream incorporation experiments.

Mutant tauRD (residues 246–408) plasmids, C-terminally fused to mEOS3.2, encoding the top 10 (alanine mutations at amino acid positions 308, 337, 346, 309, 313, 297, 310, 312, 329, and 328) and bottom 10 (alanine mutations at amino acid positions 316, 284, 317, 336, 329, 348, 345, 349, 324, and 322) were synthesized by Twist Biosciences. The virus of each construct was produced by the same lentivirus production protocol outlined above.

In order to amplify the number of aggregates in a population of cells, we treated WT tauRD-CFP cells with CBD brain lysate followed by serial rounds of cell lysis and retreatment of clarified cell lysate onto WT tauRD-CFP cells. Briefly, cells were plated in 96-well plates at 25,000 cells per well in 130 μL of media 12–16 h before treatment with clarified CBD brain lysate. Initial treatment of WT tauRD-CFP with CBD brain lysate was performed by incubating Lipofectamine 2000 (0.75 μL per well) with Opti-MEM (19.25 μL per well) for 5 min, before adding 5 μg of CBD lysate and incubating for another 30 min. After treatment, WT tauRD-CFP cells were incubated for 48 h before cell lysis was performed. Treated cells were harvested with 0.25% trypsin (GIBCO), quenched with cell media, and spun down at $100 \times g$ before being resuspended in 1XTBS buffer supplemented with cOmplete ULTRA protease inhibitor cocktail tablets, EDTA free. The resuspended cells were subjected to 3 freeze–thaw cycles. The lysates were then sonicated, centrifuged, and quantified using the same protocol used to

prepare the CBD brain lysate. The clarified cell lysate was later used as treatment onto larger well dishes (24-well plates and 6-well plates) until 60–80% of cells contained aggregates. The cells were replated into 96-well plates at 25,000 cells per well in 130 μL of media 12–16 h before treatment with 10 μL tauRD alanine mutants and control lentivirus, in triplicates.

Following 48 h of lentivirus treatment, the cells were harvested by 0.25% trypsin digestion for 5 min, quenched with cell media, transferred to 96-well U-bottom plates, and centrifuged for 5 min at 200×*g*. The cells were then fixed in PBS with 2% paraformaldehyde for 10 min, before a final centrifugation step and resuspension in 150 μL of PBS. A BD-LSR Fortessa Analyzer instrument was used to perform FRET flow cytometry analysis.

### Flow cytometry data analysis

Initial gates were made in order to screen for a population of events that were single cells and double positive for both donor fluorophore (CFP) and acceptor fluorophore (non-photoconverted mEOS3.2). FRET between fluorophores was measured by exciting cells with 405 nm violet laser, and emission was collected using a 505 nm long pass filter, and 525/50 nm band pass filter.

FCS files were exported from the FACSDiva data collection software and analyzed using FlowJo v10 software (Treestar). Compensation was manually applied to correct donor bleed-through into the FRET channel guided by a sample with non-aggregated tauRD-mEOS3.2. After selecting for single and double fluorophore positive events, samples were gated on the acceptor intensity such that cells with similar concentrations of tauRD-mEOS3.2 were analyzed to control for the contribution of variable tauRD concentrations to changes in the incorporation of alanine mutants. The gating strategy for the FRET experiments is illustrated in Supplementary Fig. 4f.

### Generation of distance matrix

To cluster the tau fibril structures, the $\Delta REU_{mut-wt}^{assembly}$ values for each residue in the fibril were first normalized, and residues not common to all fibrils in the analysis were removed. These normalized $\Delta REU_{mut-wt}^{assembly}$ values were then used to calculate a distance matrix between each pair of fibril structures using the Scipy library's 'spatial.distance_matrix' method[83].

### Structure clustering

Agglomerative clustering using Ward's method and a Euclidean distance metric was performed on the normalized $\Delta REU_{mut-wt}^{assembly}$ values obtained from the in silico alanine scan. Residues not common to all the fibrils being clustered were removed. Using the Scipy library's 'cluster.hierarchy.linkage' method, a linkage matrix describing the clustering of the structures was generated[83]. This linkage matrix was used to generate a dendrogram and plotted alongside the normalized $\Delta REU_{mut-wt}^{assembly}$ values using the Seaborn library's clustermap functionality[84,85].

### Identification of distinguishing fibril features

To identify features distinguishing fibrils, a Random Forest Classification model was generated and interpreted. The normalized $\Delta REU_{mut-wt}^{assembly}$ values for each of the thirty-five replicates obtained for each residue in the fibril were classified based on the disease of origin (AD, CBD, CTE, PiD, PSP, GGT, and GPT). Residues not found in all the structures were removed. Similar residues were grouped together using the Scikit-Learn FeatureAgglomeration functionality to recursively cluster together residues that behaved similarly in the in silico alanine scan across the different structures, using Ward's method with a Euclidean distance metric[86]. The $\Delta REU_{mut-wt}^{assembly}$ values were in this manner agglomerated to twenty clusters, each valued at the mean of the $\Delta REU_{mut-wt}^{assembly}$ values of the residues in the cluster.

This clustered $\Delta REU_{mut-wt}^{assembly}$ data was then split into a test dataset (a random 26 of the 35 replicates for each residue in the structure) and a train dataset (the remaining nine of the 30 replicates for each residue in the structure). A Random Forest Classifier with 100 classifiers was then trained on the train dataset and classification performance was calculated using the train dataset. This test-train splitting, classifier training, and scoring were repeated 2500 times and the best-scoring classifier was taken.

To identify the features being used by the classifier to distinguish the different fibril morphologies, the Python 'treeinterpreter' library was used to parse out the contribution of each cluster to the classifier's prediction of a given fibril type[87]. To do this, the $\Delta REU_{mut-wt}^{assembly}$ values produced by the in silico alanine scan for each fibril (averaged over all 35 replicates) was classified by the model, and tree interpreter was used to extract the contributions of each cluster to correct classification by the Random Forest classification model (i.e., the contribution of clusters to classification as AD for AD-PHF/SF fibrils, classification as GGT for GGT_T1/T2/T3 fibrils, etc.).

### Reporting summary

Further information on research design is available in the Nature Portfolio Reporting Summary linked to this article.

## Data availability

All ThT aggregation and $\Delta REU_{mut-wt}^{assembly}$ data for tau fibrils generated in this study are available as Source data and are also available on the Zenodo database under accession code 7549949. The raw output for the $\Delta REU_{mut-wt}^{assembly}$ data is also available on the Zenodo database under accession code 6407336. PBD id's used in this study are: 7p6d, 7p6e, 6vha, 6vh7, 6tjo, 6tjx, 5o3l, 5o3t, 6hre, 7qjw, 6nwp, 6nwq, 6hrf, 7ql4, 5o3o, 7p65, 7p66, 7p67, 7p68, 6gx5, 7p6c, 7p6a, 7p6b, 6qjh, 6qjm, 6qjp, and 6qjq. Source data are provided with this paper.

## Code availability

A protocol capture and code used for this work is made available on the UT Southwestern Gitlab instance at [https://git.biohpc.swmed.edu/s184069/flex_ddg_ala_scn_runner].

A snapshot of the code as of the time of publication can also be found on the Zenodo database under the accession code 7549915 [https://doi.org/10.5281/zenodo.7549914].

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

## Acknowledgements

L.A.J. is supported by an Effie Marie Cain Scholarship in Medical Research and a grant from the Welch Foundation (I-1928-20200401). C.L.W., M.I.D., and L.A.J. are supported by a Chan Zuckerberg Initiative (CZI) Collaborative Science Award (2018-191983). Transmission electron microscopy was performed at the Electron Microscopy Core Facility at UTSW, which is supported by the National Institutes of Health (NIH) (1S10OD021685-01A1 and 1S10OD020103-01). Computational resources were provided by the BioHPC cluster supported by the Lyda Hill Department of Bioinformatics at UTSW. We thank the members of the Joachimiak lab, particularly Sofia Bali, for the discussions and feedback on the manuscript.

## Author contributions

V.M., J.V.A., and L.A.J. initiated the project. Peptide aggregation experiments were performed by V.M. TEM of fibrils was performed by B.D.R. V.M. developed the methods for in silico minimization of fibril assemblies, estimation of the energy for WT and mutant assemblies, and data analysis. V.M. developed the methods for machine learning classification of features derived from the energies with the help of A.R.V. Validation of the tau alanine mutants in cells was performed by J.V.A. and V.B., using tauopathy patient samples collected and characterized by C.L.W. M.I.D. provided guidance on cell experiments. Finally, V.M. and L.A.J. conceived and directed the research as well as wrote the manuscript. All authors contributed to the revisions of the manuscript.

## Competing interests

The authors declare no competing interests.
