## [Peer Review File · Nature Communications]

Network of hotspot interactions cluster tau amyloid foldsEditorial Note: Parts of this Peer Review File have been redacted as indicated to maintain the confidentiality of unpublished data.

REVIEWER COMMENTS

Reviewer #1 (Remarks to the Author):

Over the last years the cryo-EM atomic structure has been solved of tau amyloid polymorphs from patients but also from in vitro heparin-induced tau amyloids. Mallapudi et al here present a detailed study analysing the differential thermodynamic stabilization of polymorphs to understand how tau can adopt alternative polymorphs. Such studies are important as they will allow us to extract the structure-activity relationships between tau conformations and specific disease phenotypes.

More specifically the authors here present an in silico alanine scanning method to estimate the energetics of amino acid interactions in 27 structural polymorphs using Rossetta. Second, they experimentally test their findings by addressing the effect of heterotypic interactions on tau amyloid assembly. Finally, they use these energetic profiles to classify amyloid polymorphs using a random forest classifier to identify distinguishing features that could in the future be manipulated to direct polymorphism.

This study is very helpful in understanding the thermodynamic make-up of tau polymorphs. The Fibril Alanine Scanning Protocol is carefully developed and the experimental validation thereof is convincing.

I have some minor concerns that I think should be addressed or discussed:

Point 1. The authors state they used CFP in the methods, but show mCerulean in the figure.

Point 2. How did the authors handle the (I expect) significantly different energetics on edge monomers, compared to internal layers for each position? And how was this positional deviation used to find key residue compared with the in cellulo data? Or for the clustering of polymorphs?

Point 3. Recently, van der Kant et al. developed a similar approach to identified thermodynamic profiles of amyloid polymorphs (van der Kant et al. 2022 Structure, Louros et al. 2021 Bioinformatics). This study should be included in the discussion. Is there a potential overlap between the different approaches? It would certainly be interesting to compare them and the resulting clustering of polymorphs.

Point 4. Cryo-EM structures often show unassigned densities suggesting additional molecules are bound to these amyloid polymorphs and which are not considered by Rossetta. Can the authors discuss whether and how this can affect their findings? E.g. could the heparin-induced fibrils then be more stable than they appear?

Point 5. VEVKSE is underlined in fig. 1 implying it is an established APR sequence, yet the experimental data provided in this study indicate it does not form ordered amyloid-like aggregates.

Point 6. The fact that several APRs predicted by ZipperDB failed to form ordered amyloid-like aggregates experimentally (all with the exception of the known PHF6 and PHF6*) maybe indicates it is not an ideal scale for amyloidogenicity? How do the other predictors mentioned perform? Interestingly, ZipperDB also predicts most PGGG turns as APR, which could be a sequence bias of ZipperDB using a single sequence model to predict topologies?

Point 8. Why are there blank spaces in the ZipperDB profile in Fig. 1? Were these segments not predicted (including VQIVYK)?

Point 9. Why are there blank spaces in the heat map in Fig 4?

Point 10. I do not understand exactly how was the dREU normalization performed? There was a 5-window strategy used? Did the authors consider the residue position in different layers? What exactly was used in each step of the analysis?

Reviewer #2 (Remarks to the Author):

This manuscript reports an in silico analysis of the interactions that stabilize amyloid fibrils. The central idea is that computational methods can be used to estimate the change in the stability of the fibrils upon mutations, as well as the network of interactions that stabilize the fibrils themselves.

The authors report a detailed in silico mutational analysis of amyloidogenic sequence motifs of tau. Their conclusions highlight the importance of non-polar interactions in stabilising amyloidogenic motifs within fibril structures. They also put forward the suggestion that different fibril polymorphs may arise from different arrangements of these motifs upon chemical changes, such as mutations or post-translational modifications.

Although the in silico alanine scanning approach is elegant and informative, it is limited by the assumption that the chemical changes should be sufficiently small to conserve the overall fold of the fibril polymorph. Since the structure of tau fibrils has been shown to sensitive to perturbations, this assumption may be problematic. This could be particularly the case concerning the authors' suggestion that their approach is going to help understand the role of pathogenic mutations or post-translational modifications of amyloidogenic proteins.

To demonstrate this last point, the cell-based aggregation experiments are crucial, so they should be explained in more detail. In addition, to demonstrate impact, the authors should show that the results in Figure 4f cannot be obtained using existing sequence-based method of predicting aggregation propensities.

It would also be important to compare the network of interactions that stabilize a fibril structure determined through the mutational approach presented here, with the corresponding network obtained from a more straightforward approach based on the analysis of the interaction strengths of the wild-type sequence in that fibril structure. This second approach would seem more robust because it is not affected by perturbations of the structure. In this sense, the authors should comment on the possible alternative implementation of their overall research plan through AlphaFold, which is also capable, at least to some extent, to predict the structure of an amyloid fibril from the sequence of a protein.

Again to demonstrate impact, the significance of the classification of tauopathies in Figure 6 should be clarified. The authors should explain better how this classification help rationalize our current understanding of these different diseases.

Reviewer #3 (Remarks to the Author):

The authors introduce an in silico alanine scan method to assess the contributions of amino acid stretches to amyloid fibril formation. The method was applied to tau and 27

polymorphisms. Predictions are tested in vitro and in cell. A machine learning approach is presented to classify diverse tau fibril structures and related interaction motifs.

I particularly liked the analysis of energetics, and especially the Δ SASA (single layer, fibril) interpretation of the role of hydrophobic residues. Simple and straightforward.

Major comments.

I find missing the link between fibrils' polymorphisms and pathology / severity of diseases.

For instance, what is unique of heparin derived fibril structures? Could the author speculate on the meaning of their results (highest energy) in the cellular context?

I also think that it would be great if the authors made the manuscript less technical (details can be still in the Materials and Methods section) and discussed more the potential for applications.

The discussion is a little bit a summary of the results and should be expanded.

Specific points.

Some aspects could be put in context, for instance "we developed an in silico alanine scan method"...it is a standard approach since very long time, the authors should emphasize that they manage to mirror experimental approaches with their computational method...

The authors test their in silico predictions using in vitro experiments on hexapeptides, and especially 306-VQIVYK-311.

Yet, in the first part of the results the authors make a point that some residues participate in the formation of the fibril although they are not in the hydrophobic core. This suggests that entire regions, in addition to the hexapeptides in the hydrophobic core should be considered. This part should be better discussed and linked with the preeminent role of 306-VQIVYK-311. In the context of heterotypic interactions, I particularly like that the peptide 350-VQSKIG-355 completely block the aggregation of 306-VQIVYK-311, which could even have therapeutic potential.

For the broad audience, I would introduce Rosetta and how the authors used it. The description is too short for me.

Towards the end of the results, the authors make an important point that their method can help to identify structural differences that cannot be revealed by antibodies. Yet, there is no discussion on this and why the method is much better than others.

In the discussions, the authors mentioned that "distinct tau fibrils are associated with difference disease": it should be useful if they expanded on how the energy of the different mutations could explain some aspects of tauopathies (seeding/spreading/morphology)

We are very grateful to the reviewers for their critical and detailed evaluation of our study. We also thank the reviewers for their positive assessment of our work and helpful suggestions. Please find below our point-by-point response addressing reviewer comments in full to improve and streamline the final manuscript. We hope that the reviewers will find the revised version of the manuscript suitable for publication in Nat Comm.

REVIEWER COMMENTS

Reviewer #1 (Remarks to the Author):

Over the last years the cryo-EM atomic structure has been solved of tau amyloid polymorphs from patients but also from in vitro heparin-induced tau amyloids. Mullapudi et al here present a detailed study analysing the differential thermodynamic stabilization of polymorphs to understand how tau can adopt alternative polymorphs. Such studies are important as they will allow us to extract the structure-activity relationships between tau conformations and specific disease phenotypes.

More specifically the authors here present an in silico alanine scanning method to estimate the energetics of amino acid interactions in 27 structural polymorphs using Rosetta. Second, they experimentally test their findings by addressing the effect of heterotypic interactions on tau amyloid assembly. Finally, they use these energetic profiles to classify amyloid polymorphs using a random forest classifier to identify distinguishing features that could in the future be manipulated to direct polymorphism.

This study is very helpful in understanding the thermodynamic make-up of tau polymorphs. The Fibril Alanine Scanning Protocol is carefully developed and the experimental validation thereof is convincing.

I have some minor concerns that I think should be addressed or discussed:

Point 1. The authors state they used CFP in the methods, but show mCerulean in the figure.

We thank the reviewers for pointing this out. CFP tagged tauRD was indeed used in the cell experiments to form the initial intracellular CBD-derived inclusions as was described in the methods. We have corrected Figure 4i.

Point 2. How did the authors handle the (I expect) significantly different energetics on edge monomers, compared to internal layers for each position? And how was this positional deviation used to find key residue compared with the in cellulo data? Or for the clustering of polymorphs?

We thank the reviewers for this wonderful suggestion. We had originally noticed an exaggerated edge contribution in our assemblies in the context of our analysis of the $\Delta\Delta G_{mut-wt}^{binding}$ comparing the unbound vs bound states (Supplementary Fig. 4b-d). We considered including this analysis for the assembly energetics ($\Delta REU_{mut-wt}^{assembly}$) in the original manuscript but decided against it as

the edges only contributed nominally to the total score and that the total energy change ($\Delta\text{REU}_{\text{mut-wt}}^{\text{assembly}}$) correlated strongly with the energy change in only the internal ($\Delta\text{REU}_{\text{mut-wt}}^{\text{internal}}$) layers as a function of total number of layers. Motivated by the reviewers' comments, we have included this analysis in the Supplementary Figures (Supplementary Fig. 4e-k) and described these analyses on page 14 of our revised manuscript. In our multi-layer calculations for the AD-PHF and CBD fibrils (n=3 to n=9), we computed the total ($\Delta\text{REU}_{\text{mut-wt}}^{\text{assembly}}$), the internal layers ($\Delta\text{REU}_{\text{mut-wt}}^{\text{internal}}$) and the edge layers ($\Delta\text{REU}_{\text{mut-wt}}^{\text{edge}}$) scores and compared the energetics of each. We find that the total assembly ($\Delta\text{REU}_{\text{mut-wt}}^{\text{assembly}}$) and the internal layer ($\Delta\text{REU}_{\text{mut-wt}}^{\text{internal}}$) per residue changes in energies correlate positively across each layer analysis for both AD-PHF and CBD (see panels a and b below, and also Supplementary Fig. 4e-i). Comparison of the $\Delta\text{REU}_{\text{mut-wt}}^{\text{internal}}$ and $\Delta\text{REU}_{\text{mut-wt}}^{\text{edge}}$, the energetics yield a poorer correlation but still show an overall negative relationship between these datasets indicating that the mutations to alanine in edge layers leads to stabilization which we attribute to changes in the solvation energies of predominantly larger nonpolar residues that are solvent exposed, while alanine mutations to those positions in the internal layers lead to destabilization due to formation of voids. Consistent with this observation, residues that are solvent exposed (i.e. face outward) in a fibril, often have small changes in energetics in response to mutations (i.e. are not making interactions) and are independent of layer (i.e. edge vs internal) because they are similarly solvent exposed. Finally, the Pearson correlation coefficient comparing the $\Delta\text{REU}_{\text{mut-wt}}^{\text{assembly}}$ and $\Delta\text{REU}_{\text{mut-wt}}^{\text{internal}}$ energetics yields a strong positive correlation (>0.9) while the edge and internal layers yield a weaker correlation that changes with the number of layers (see panel c below and also Supplementary Fig. 4j, k). We interpret this in the context of the RMSD changes of the minimized assemblies to the input structure and how they decrease (and the energetics improve) with increasing numbers of layers but that this is also dependent on the monomer fold (i.e., AD PHF and CBD minimize differently). For example, in the 3-mer assemblies, there is a much smaller distinction between the internal vs edge layers while in the larger assemblies the core dominates the overall packing which is more constrained. Thus using larger assemblies yield structures more similar to the input (by RMSD) in which the overall energetics of the assembly are dominated by the internal layers. We are currently implementing this approach on a larger set of different fibrils and will include $\Delta\text{REU}_{\text{mut-wt}}^{\text{internal}}$, $\Delta\text{REU}_{\text{mut-wt}}^{\text{edge}}$, and $\Delta\text{REU}_{\text{mut-wt}}^{\text{internal}}$ in future analyses. We are unsure how these relationships will change in fibrils with proteins that have different amino acid compositions.

Finally, we compare the top hits for $\Delta\text{REU}_{\text{mut-wt}}^{\text{assembly}}$ that we experimentally tested for the CBD fibril (Fig. 4g-i) and compare them to the top hits identified using the $\Delta\text{REU}_{\text{mut-wt}}^{\text{internal}}$ energetics. We find that 8 amino acid positions in the CBD fibril are common between $\Delta\text{REU}_{\text{mut-wt}}^{\text{assembly}}$ and $\Delta\text{REU}_{\text{mut-wt}}^{\text{internal}}$ (see panel d below, top). Importantly, the hotspot hits missing in the top 10 are within the top 12 of each dataset. Furthermore, in the top 20 hits we see 17 residues that are common (see panel d below, bottom). Overall, the $\Delta\text{REU}_{\text{mut-wt}}^{\text{assembly}}$ and $\Delta\text{REU}_{\text{mut-wt}}^{\text{internal}}$ have very high congruence and the nonpolar residues that stabilize VQIVYK and VEVKSE interactions are present at the top of both datasets in of the CBD fibril analysis. Thus, both analyses yield the

same hits. Regarding the machine learning approach, because the rank order and patterns are very similar between the $\Delta REU_{mut-wt}^{internal}$ vs $\Delta REU_{mut-wt}^{assembly}$ per residue energetics we suspect that we would identify similar features that distinguish the structures. For example, the D358A mutation in both $\Delta REU_{mut-wt}^{internal}$ and $\Delta REU_{mut-wt}^{assembly}$ of the CBD structure analysis are both strong hits (3/10 and 6/10, respectively) and therefore this feature should be preserved. The comparison of the total, internal and edge layer energetics is described on pages 13-14 and the methods.

Recently, van der Kant et al. developed a similar approach to identified thermodynamic profiles of amyloid polymorphs (van der Kant et al. 2022 Structure, Louros et al. 2021 Bioinformatics). This study should be included in the discussion. Is there a potential overlap between the different approaches? It would certainly be interesting to compare them and the resulting clustering of polymorphs.

We thank the reviewers for this great suggestion, we have included a mention of the work from van der Kant et al. 2022 Structure, Louros et al. 2021 Bioinformatics in the introduction (page 5) as well a brief comparison of our work to this recently published work in the discussion (page 24).

Point 4. Cryo-EM structures often show unassigned densities suggesting additional molecules are bound to these amyloid polymorphs and which are not considered by Rosetta. Can the authors discuss whether and how this can affect their findings? E.g. could the heparin-induced fibrils then be more stable than they appear?

We thank the reviewers for this suggestion. As astutely observed by the reviewer, we found that in the heparin-induced fibril structures the per residue contribution is smaller than in the disease fibril structures which is also consistent with the fibril structures not burying as much surface area – the cores are visibly less packed. An important question is therefore: how are these fibril structures formed if the stability is much weaker? As the reviewer suggests it is possible that heparin binding to the fibril surfaces could stabilize the fibrils, but this is difficult to conclude definitively as the electron density for the bound heparin in the cryo-EM structures is weak, likely due to heparin binding heterogeneity. Thus, it is likely that heparin contributes to the overall stability of the heparin-induced fibrils but without a clear structure of fibril bound to heparin it will be difficult to ascertain with confidence how it contributes to stability. Similarly, in the CBD and CTE structures additional densities thus residues that contact these unknown molecules likely would be further stabilized. We have included a description of these ideas in the results on pages 11 and in the discussion on page 27.

Point 5. VEVKSE is underlined in fig. 1 implying it is an established APR sequence, yet the experimental data provided in this study indicate it does not form ordered amyloid-like aggregates.

We thank the reviewers for noticing this. Indeed, ZipperDB predicts that VEVKSE should be compatible with fibril formation, but it does not aggregate under our conditions. We have removed the underline under VEVKSE in Fig. 1.

Point 6. The fact that several APRs predicted by ZipperDB failed to form ordered amyloid-like aggregates experimentally (all with the exception of the known PHF6 and PHF6*) maybe indicates it is not an ideal scale for amyloidogenicity? How do the other predictors mentioned perform? Interestingly, ZipperDB also predicts most PGGG turns as APR, which could be a sequence bias of ZipperDB using a single sequence model to predict topologies?

This is a great suggestion. We have now calculated amyloidogenicity of the relevant tau fragment that spans the fibril cores using ZipperDB, Waltz and Pasta. The consensus across all three algorithms is that VQIVYK and VQIINK are strong aggregating peptides (see below, Fig. 1). All three algorithms also predict that VQISKIGS (and portion of following sequence) would aggregate. Additionally, both Pasta and ZipperDB algorithms, predict several other sequences but none of these other sequences aggregated in the conditions that we tested. We have included this analysis in Supplementary Fig. 1a to highlight that all the algorithms are in agreement that VQIVYK and VQIINK are core aggregating peptides and that predictions from Waltz are the most accurate. In future experiments, it would be really interesting to further optimize the aggregation conditions of this fragment to understand how this sequence may aggregate on its own beyond the observation that this sequence is an efficient suppressor of VQIVYK aggregation.

Point 8. Why are there blank spaces in the ZipperDB profile in Fig. 1? Were these segments not predicted (including VQIVYK)?

We thank the reviewer for noticing these gaps. These gaps are present because the ZipperDB algorithm relies on structure-based threading of the query sequence followed by minimization in Rosetta to evaluate the energetics of the assembly. If the sequence window includes proline in the middle of the fragment, the energetics will be penalized because it is unable to form backbone hydrogen bonds at this site but also changes the conformation of the backbone which leads to steric clashes thus yielding positive energies in the analysis. We had added a brief description of why there are gaps in the methods section under “*Amyloid motif prediction in tau and in vitro ThT peptide fibrillization*” (page 28).

Point 9. Why are there blank spaces in the heat map in Fig 4?

We thank the reviewers for pointing this out and apologize for not more clearly describing why glycine positions were omitted from our final energetic interpretations. A more detailed analysis of the glycine torsional angles in the fibril structures uncovered that glycine residues adopt unique torsional angles that deviate from normal distributions of high-resolution structures of proteins observed in the Protein Data Bank (PDB). We compared the torsional angle distributions of cryo-EM tau fibrils (determined using helical reconstruction in Relion), all cryo-EM fibrils (determined using helical reconstruction in Relion) to distributions of high-resolution structures of globular proteins from the PDB revealed nuanced differences that appear to be enriched/abundant in fibrils (Supplementary Fig. 3g-i). We additionally found that glycine residues in cryo-EM fibril structures (also tau) adopt torsional angles outside of ranges typically observed for alanine. Furthermore, in our alanine scan approach we found that placement of alanine residues at these glycine sites led to phi-psi penalties and VdW clashes that led to significant destabilization (see Figure below and Supplementary Figure 3e,f) that did not inform us of interactions that could guide stability of the structures beyond that glycines are important. We thus decided to not include the alanine mutations at glycine sites in our profiles and focused on sites that give more easily interpretable results such as energetics of side-chain to side-chain interactions. The justification for this rationale is described in the results on page 12, methods on pages 30-31 and the new analysis is shown in Supplementary Figure 3e-i.

Point 10. I do not understand exactly how was the dREU normalization performed? There was a 5-window strategy used? Did the authors consider the residue position in different layers? What exactly was used in each step of the analysis?

We thank the reviewer for pointing out a need to clarify how the normalization was carried out and what was calculated at each step. Our method calculates the difference in energy $\Delta REU_{mut-wt}^{assembly}$ between the sum total of a WT and a mutant assembly in which a single amino acid position is mutated across all layers (see Fig 3a). For AD PHF and CBD we compared assemblies ranging from $n=3$ to 9 layers. All subsequent calculations on the remaining fibrils were carried out with $n=9$ layers. Regarding normalization, within each fibril dataset the maximal signal from $\Delta REU_{mut-wt}^{assembly}$ was normalized to 100 units. These values were mapped onto each structure in Fig. 4c to illustrate that buried residues are more important in the fibrils. Additionally, in Fig. 4b, we also computed normalized values for windows of 5 valid residues (excluding glycine positions) to allow a direct comparison to the amyloid motif prediction in ZipperDB to identify motifs that stabilize the interactions rather than single amino acids.

Reviewer #2 (Remarks to the Author):

This manuscript reports an *in silico* analysis of the interactions that stabilize amyloid fibrils. The central idea is that computational methods can be used to estimate the change in the stability of the fibrils upon mutations, as well as the network of interactions that stabilize the fibrils themselves.

The authors report a detailed *in silico* mutational analysis of amyloidogenic sequence motifs of

tau. Their conclusions highlight the importance of non-polar interactions in stabilising amyloidogenic motifs within fibril structures. They also put forward the suggestion that different fibril polymorphs may arise from different arrangements of these motifs upon chemical changes, such as mutations or post-translational modifications.

Although the *in silico* alanine scanning approach is elegant and informative, it is limited by the assumption that the chemical changes should be sufficiently small to conserve the overall fold of the fibril polymorph. Since the structure of tau fibrils has been shown to be sensitive to perturbations, this assumption may be problematic. This could be particularly the case concerning the authors' suggestion that their approach is going to help understand the role of pathogenic mutations or post-translational modifications of amyloidogenic proteins.

Alanine scans have been commonly employed to evaluate folding of proteins and protein-protein interactions. Our original motivation for employing an alanine scan rather than capturing the energetics of residues *in place* was to develop a framework for the design of fibril structures and secondly to begin testing the contribution of different residues which is only possible with mutagenesis. This is also highlighted by our efforts to validate our observations which relied on the usage of mutants in *in vitro* peptide and cell-based to validate our observations (Fig. 4 and Supplementary Fig. 4). Excitingly, the cell experiments capture the differences we predict from the simulations that highlight specific residues to be important for assembly of a specific tau fibril (i.e. hotspot) and conversely identify positions that are dispensable (i.e. neutral) (see Fig. 4). This duality of amino acid contributions may be an exciting avenue to begin to discriminate different fibril structures that is currently being explored in the lab.

We are excited that we have now developed an *in silico* mutagenesis approach as the basis for protein design of fibrils and are moving forward to create new tau sequences designed towards specific tau folds. We predict that we may preferentially be able to adopt the designed fold conformations (i.e. positive design) and in parallel optimize the sequences that would explicitly disfavor formation of other folds (i.e. negative design). We have begun employing this approach to modify a local cluster of interactions informed by an FTD-tau pathogenic mutation that lead to spontaneous tau aggregation *in vitro* and in cells yielding designed tau sequences that can themselves aggregate spontaneously (Chen et al bioRxiv 2022, doi.org/10.1101/2022.08.11.503511). We are in the process of validating these designed fibrils using cryo-EM to test whether they indeed adopt the desired fibril conformation.

In the context of post-translational modifications such as phosphorylation these often lie outside of the fibril core, thus may influence the dynamics of the flanking N- and C-termini to promote uncovering of the repeat domain. It remains unknown how PTMs may contribute to the formation of tau structural polymorphs. In terms of pathogenic mutations – our method indeed can be used to ascertain whether a mutation is compatible with a given structural scaffold and we have used experiments to validate this observation. Using our mutagenesis strategy, we have screened all known FTD-tau linked disease mutations on AD-PHF and CBD fibril conformations and discovered that the P301L mutation destabilizes the CBD conformation and has no effect on AD-PHF because it falls outside of the fibril core (see panel a below).

Consistent, with these predictions AD tauopathy material seeds 4-fold better on cells expressing tauRD P301L compared to CBD indicating that our methods indeed can uncover sites that disrupt folding of tau into specific conformations to propagate a certain tau amyloid fold (see panel b below).

[Redacted]

In summary, the reviewer brings up an excellent point that structures are sensitive to mutations but importantly there is a pattern of sensitivity that makes physical sense based on the interactions. Buried nonpolar interactions are important while surface positions are not. We thus can leverage these different sites to probe structure and gain insight into how to control folding these conformations and to begin designing possible strategies to selectively diagnose these different conformations.

To demonstrate this last point, the cell-based aggregation experiments are crucial, so they should be explained in more detail. In addition, to demonstrate impact, the authors should show that the results in Figure 4f cannot be obtained using existing sequence-based method of predicting aggregation propensities.

We thank the reviewer for this suggestion. We have expanded on the description of the cell-based experiments in the results (page 19) that begin to validate our approach as well as deeper description of how this assay may be useful as a diagnostic tool in the discussion (page 27). In terms of using simple sequence-based methods these will likely fail as our data suggest that the VQIVYK motif interacts with other non-amyloidogenic sequences that present nonpolar residues. Perhaps, what complicates this interpretation is that some of these other sequences are predicted to form fibrils but experimentally they cannot. The only two sequences that stably and reproducibly form fibrils in our hands using relatively standard conditions are VQIVYK and VQIINK (and are consensus hits across the three tested aggregation prediction algorithms) and while others are predicted to form amyloids further experiments would be required to confirm this. Thus, amyloid predictions are only accurate for high scoring hits and lower tier hits may assemble but would require further testing. As a result, it is difficult to infer more information from sequence-based methods alone.

For example, in our CBD hotspot analysis, the interactions between VQIVYK and VEVKSE are predicted to be important. Only the VQIVYK sequence forms fibrils while VEVKSE forms

amorphous aggregates. It is not clear why the VEVKSE is unable to self-assemble but it is likely that a glutamic acid is a gatekeeper residue and this is not correctly captured in the aggregation propensity predictions – perhaps performing the reaction in low pH could help but then this would potentially be non-physiological. Furthermore, what complicates efforts to predict what residues are important is that tau adopts a diversity of structures that while all use VQIVYK, the interactions with VQIVYK vary and involve sequences that contain nonpolar character but are not explicitly aggregation prone. We suspect that the VQIVYK sequence can nucleate interactions with other nonpolar residues but that these interacting sequences are not absolutely required to be amyloidogenic but rather only need to present a nonpolar pattern that is compatible with VQIVYK.

It would also be important to compare the network of interactions that stabilize a fibril structure determined through the mutational approach presented here, with the corresponding network obtained from a more straightforward approach based on the analysis of the interaction strengths of the wild-type sequence in that fibril structure. This second approach would seem more robust because it is not affected by perturbations of the structure. In this sense, the authors should comment on the possible alternative implementation of their overall research plan through AlphaFold, which is also capable, at least to some extent, to predict the structure of an amyloid fibril from the sequence of a protein.

This is a great point, and this is likely possible but our motivation of replacing residues with alanine is essential for experimental validation and our future interests of designing tau sequences with new features (see above). As suggested by the reviewer we have compared the in place per residue energetics of the minimized fibril structures (summed over the chains in the assembly) and compared them to our calculated $\Delta\text{REU}_{\text{mut-wt}}^{\text{assembly}}$. The correlation coefficient for AD-PHF and CBD structures is -0.69 and -0.62, respectively (see figure below). As expected, there is a negative correlation which captures the energetics of the interaction of the WT amino acid (should be negative if important) to our calculated $\Delta\text{REU}_{\text{mut-wt}}^{\text{assembly}}$ where positive values identify amino acids that when mutated to alanine are destabilizing. We again show our top 10 hits identified in $\Delta\text{REU}_{\text{mut-wt}}^{\text{assembly}}$ (see below, right panel, colored in red) and we see that 8 of the 10 amino acids are predicted correctly. This comparison shows overall similarity in the two metrics and is more related to recent work from Van der Kant *et. al.* in which they computed per residue energetics for different fibrils in place, also highlighting congruency between Rosetta and FoldX (PMID 35609599).

Regarding, prediction of structures from sequence alone using machine learning similar to AlphaFold – this is certainly a possibility but at this stage there do not exist sufficient numbers of structures (nearly a 1/3 of all fibril structures are also of tau) from diverse proteins to train on yet but we anticipate as the number of structures increase this may be possible. Furthermore, unlike globular proteins which most often fold into a single conformations which is the global minimum many of these intrinsically disordered proteins encode the capacity to adopt a set of different structures (i.e. PrP, Ab, alpha-synuclein, TDP-43, FUS, IAPP, tau etc.), so the methods would need to build ensembles of different possible structures defined by the common rules that minimally may be represented by all possible heterotypic interactions with amyloidogenic motifs. We have included a discussion about the challenges and future application of machine learning methods as applied to prediction of fibril structures on page 27.

Again to demonstrate impact, the significance of the classification of tauopathies in Figure 6 should be clarified. The authors should explain better how this classification help rationalize our current understanding of these different diseases.

We thank the reviewer for this suggestion. The prior knowledge for how the fibrils were classified were dictated predominantly by the presence of certain repeats in the fibril conformation (PMID 34588692). Here we use the energetic contribution of each amino acid to classify the structures. We identify features that are important for all structures but importantly we also identify sites that are differentially used to distinguish the structures. We have expanded on the significance of these results in the discussion (page 26).

Reviewer #3 (Remarks to the Author):

The authors introduce an in silico alanine scan method to assess the contributions of amino acid stretches to amyloid fibril formation. The method was applied to tau and 27 polymorphisms. Predictions are tested in vitro and in cell. A machine learning approach is presented to classify diverse tau fibril structures and related interaction motifs.

I particularly liked the analysis of energetics, and especially the Δ SASA (single layer, fibril) interpretation of the role of hydrophobic residues. Simple and straightforward.

Major comments.

I find missing the link between fibrils' polymorphisms and pathology / severity of diseases.

We thank the reviewer for pointing this out. Tauopathies encompass over 25 different diseases where tau deposits as amyloids in human brain. For many years neuropathologists have described the morphology, cell type and localization of the tau deposits as hallmarks of each different tauopathy – these data hinted that the tau fibril conformations across these diseases may not be the same. Early biochemical work suggested that these conformations may be distinct and can propagate in a prion-like manner. The emergence of ex vivo cryo-EM structures from patient-derived samples for the first time validated these early observations in a rather convincing fashion that indeed the changes in gross tau deposit morphology in tissues indeed translates to unique protein conformations. Regarding to severity, it is not clear what the relationship of structure to disease severity. It is important to note that clinical presentation of disease can overlap for different pathologies suggesting that deposition of a protein aggregate in a localized brain region can manifest in specific symptoms independent of pathology – a good example of this is frontotemporal dementia – deposition of amyloids of many proteins, including tau (i.e. FTD-tau) are linked to this clinical syndrome and are unified by degeneration of the frontal cortex. So while it's not clear how structure relates to severity, the location of the aggregates in brain regions and cell types is linked to the clinical presentation. Severity may be linked to capacity to replicate a specific conformation of tau fibrils in a prion-like manner, but this remains to be well understood. We have included a discussion paragraph “Structural polymorphs and disease” on page 24 that highlights how structural knowledge of tau fibrils from disease may help us understand disease mechanisms but also propose novel strategies for diagnosis and therapeutics geared towards conformation-specific detection of disease.

For instance, what is unique of heparin derived fibril structures? Could the author speculate on the meaning of their results (highest energy) in the cellular context?

In the paper that describes the heparin-derived structures (PMID 30720432), the authors observed a diversity of structures in one fibril preparation. Furthermore, the heparin-derived structures were all quite different from the disease conformations. From the energetics, it appears that the heparin structures are perhaps less stable and overall bury less surface area but nonetheless studies from Diamond *et. al.* have shown that it is possible to stably propagate heparin-derived conformations of tau aggregates in cell models (Sanders et al Neuron 2014, PMID 24857020). Reviewer #1 pointed to a possibility that the heparin may be loosely bound to the fibrils thus contributing to the energetics but disordered enough to not be resolvable by averaging methods such as cryo-EM. We have included a brief comment in the results on page 11 for why the heparin-derived structures may have higher energy per residue compared to the disease-derived conformations.

I also think that it would be great if the authors made the manuscript less technical (details can be still in the Materials and Methods section) and discussed more the potential for applications.

We have moved statements from the results into the methods to help streamline the narrative. We have also expanded the discussion to describe in more detail potential application of our method (pages 25 and 27).

The discussion is a little bit a summary of the results and should be expanded.

We thank the reviewers for pointing out the overlap between the results and discussion. We have now expanded the discussion to include suggestions from below.

Specific points.

Some aspects could be put in context, for instance "we developed an in silico alanine scan method"...it is a standard approach since very long time, the authors should emphasize that they manage to mirror experimental approaches with their computational method...

Great point – we have now acknowledged the long history of implementing both experimental and in silico alanine scans to probe the importance of residues in protein folding and protein-protein interactions (page 9).

The authors test their in silico predictions using in vitro experiments on hexapeptides, and especially 306-VQIVYK-311.

Yet, in the first part of the results the authors make a point that some residues participate in the formation of the fibril although they are not in the hydrophobic core. This suggests that entire regions, in addition to the hexapeptides in the hydrophobic core should be considered. This part should be better discussed and linked with the preeminent role of 306-VQIVYK-311. In the context of heterotypic interactions, I particularly like that the peptide 350-VQSKIG-355 completely block the aggregation of 306-VQIVYK-311, which could even have therapeutic potential.

This is a great point and in some way brings it back to model systems we have developed in our lab (PMID 31175300 and 29988016) in which we think local contacts with the amyloid motif are regulatory and prevent the formation of these longer range fold determining conformations. Furthermore, we think that the combinations of how these different elements are exposed are the initial steps towards assembly of distinct structural polymorphs. We have included a discussion of this on pages 25-27.

For the broad audience, I would introduce Rosetta and how the authors used it. The description is too short for me.

We thank the reviewers for this great point – we have included a more expanded description of Rosetta in the results section and the methods (pages 9 and 30).

Towards the end of the results, the authors make an important point that their method can help to identify structural differences that cannot be revealed by antibodies. Yet, there is no discussion on this and why the method is much better than others.

We apologize for not clarifying this point. From our data, some residues (i.e., VQIVYK) are important for all fibril structures but others differentially contribute to stability across the structures, therefore, providing sensitivity to discriminate between different fibril conformations. One example derived from the machine learning classification of the energy profiles is D358 which makes stabilizing contacts in the CBD/AGD structures but in the other structures it faces outward and thus should be dispensable for folding. Experimental validation of all of these specificity determining sites would be required to test these predictions in more detail. We have included a discussion paragraph to describe how these methods can potentially be leveraged to diagnose tauopathies based on tau fibril conformation that are unique from biologics that would simply bind to aggregates and presumably clear them (page 24).

In the discussions, the authors mentioned that "distinct tau fibrils are associated with difference disease": it should be useful if they expanded on how the energy of the different mutations could explain some aspects of tauopathies (seeding/spreading/morphology)

This is a very interesting point. It is not clear what features of a fibril structure are required for its prion-like propagation. A priori, one might assume that stability of a fold may be important but in the prion protein field, more virulent Prp strains are thought to be more labile so its not clear how stability or fold determines the capacity of the structure to propagate. From our cell experiments, we may be able to infer what residues are important for presumably adopting the conformation of a monomer for it to incorporate into an existing fibril suggesting that these networks of hotspots may indeed play a role in propagation of a conformation. A corollary to this observation is that these networks of interactions are important for a tau monomer to adopt conformations similar to the fibril to incorporate our cellular validation experiments suggest that mutations can prevent incorporation of the monomer into a fibril. This might indicate that the conformations that the monomer can sample are essential steps for the initial species that are on pathway to a specific fibril conformation. We have expanded additional discussion of these concepts in the discussion (pages 25-26).

REVIEWERS' COMMENTS

Reviewer #1 (Remarks to the Author):

The authors have carefully and adequately addressed my comments.
I have no further remarks and would be satisfied with publication of the current version of the manuscript

Reviewer #2 (Remarks to the Author):

The authors have convincingly addressed all the points that I have made in my original report.

Reviewer #3 (Remarks to the Author):

I am satisfied with authors' response and I think the work should be published in Nat. Comm.

We thank the reviewers for their positive feedback on our revised manuscript and are excited that all reviewers agree that the manuscript is ready for publication in Nature Communications.

REVIEWERS' COMMENTS

Reviewer #1 (Remarks to the Author):

The authors have carefully and adequately addressed my comments. I have no further remarks and would be satisfied with publication of the current version of the manuscript

We thank the reviewers for their positive assessment of our work.

Reviewer #2 (Remarks to the Author):

The authors have convincingly addressed all the points that I have made in my original report.

We thank the reviewers for their positive assessment of our work.

Reviewer #3 (Remarks to the Author):

I am satisfied with authors' response and I think the work should be published in Nat. Comm.

We thank the reviewers for their positive assessment of our work.